# Extent and reproduction of coastal species on plastic debris in the North Pacific Subtropical Gyre

Linsey E. Haram [1] ✉, James T. Carlton[2], Luca Centurioni[3], Henry Choong [4], Brendan Cornwell[5], Mary Crowley[6], Matthias Egger[7], Jan Hafner[8], Verena Hormann [3], Laurent Lebreton[7], Nikolai Maximenko[8], Megan McCuller[9], Cathryn Murray[10], Jenny Par[1], Andrey Shcherbina[11], Cynthia Wright[10] & Gregory M. Ruiz [1]

We show that the high seas are colonized by a diverse array of coastal species, which survive and reproduce in the open ocean, contributing strongly to its floating community composition. Analysis of rafting plastic debris in the eastern North Pacific Subtropical Gyre revealed 37 coastal invertebrate taxa, largely of Western Pacific origin, exceeding pelagic taxa richness by threefold. Coastal taxa, including diverse taxonomic groups and life history traits, occurred on 70.5% of debris items. Most coastal taxa possessed either direct development or asexual reproduction, possibly facilitating long-term persistence on rafts. Our results suggest that the historical lack of available substrate limited the colonization of the open ocean by coastal species, rather than physiological or ecological constraints as previously assumed. It appears that coastal species persist now in the open ocean as a substantial component of a neopelagic community sustained by the vast and expanding sea of plastic debris.

Rafting, or the association of organisms with floating debris[1], has been an inferred mode of marine species dispersal since the nineteenth century[2]. Yet empirical evidence of floating debris' role in long-term, transoceanic rafting of coastal marine species is limited. The importance of coastal species dispersal by open-ocean rafting may depend largely on the nature of the raft material[3]. Natural rafts consist of buoyant, floating vegetation or pumice (the buoyant rock formed during volcanic eruptions). Natural materials are relatively short lived, decomposing at sea over a matter of months or a few years, becoming waterlogged and sinking, or being biodegraded or consumed by marine animals[2,4,5].

Anthropogenic materials also act as ocean rafts. Ephemeral anthropogenic materials, such as lumber, glass and metal, are made of naturally occurring materials and may not last at sea[6]. However, enduring plastic materials may survive much longer, although degradation rates vary across polymer type, habitat and environmental conditions[7–10]. Floating plastic materials, such as buoys and floats, built to persist in harsh marine environments, are by nature more durable and buoyant than natural materials, making floating plastics optimal rafts for long-distance and long-term dispersal.

Transoceanic dispersal on rafts is a high-risk, low-probability endeavour for coastal organisms, such that oceans typically represent

[1]Smithsonian Environmental Research Center, Edgewater, MD, USA. [2]Coastal & Ocean Studies Program, Williams College and Mystic Seaport Museum, Mystic, CT, USA. [3]Scripps Institution of Oceanography, University of California, San Diego, La Jolla, CA, USA. [4]Royal British Columbia Museum, Victoria, British Columbia, Canada. [5]Hopkins Marine Station, Stanford University, Pacific Grove, CA, USA. [6]Ocean Voyages Institute, Sausalito, CA, USA. [7]The Ocean Cleanup Foundation, Rotterdam, The Netherlands. [8]International Pacific Research Center, School of Ocean & Earth Science & Technology, University of Hawaii at Manoa, Honolulu, HI, USA. [9]North Carolina Museum of Natural Sciences, Raleigh, NC, USA. [10]Institute of Ocean Sciences, Fisheries & Oceans Canada, Sidney, British Columbia, Canada. [11]Applied Physics Laboratory, University of Washington, Seattle, WA, USA. ✉e-mail: linseyharam@gmail.com

major dispersal barriers and biogeographic boundaries for species distributions. The entrainment of biofouled rafted materials in ocean currents can result in several different outcomes: (1) sinking offshore, (2) exposure to inhospitable conditions leading to mortality, (3) remaining adrift in ocean currents or gyres, (4) landing in an inhospitable environment or (5) landing in a hospitable insular or continental environment. Predation at sea on biofouling may occur during scenarios 2 and 3. Until now, scenario 5 was considered the only successful oceanic rafting outcome. Indeed, while successful shorter-distance (<2,000 km) rafting has been recorded[2,11–13], successful long-distance, transoceanic dispersal of coastal organisms on either natural or anthropogenic rafts resulting in continental landing has rarely been observed and documented[14].

An example of one such rare event occurred when millions of objects were swept into the North Pacific Ocean due to the Great East Japan Tsunami of March 2011. In 2012, tsunami debris with living Japanese species began washing ashore in North America and the Hawaiian Islands. By 2015, at least 100,000 tsunami debris items landed in North America[15]. Carlton et al.[16,17] and Hansen et al.[18] reported that 381 living Japanese coastal species landed in North America and Hawaii between 2012 and 2017 primarily on plastic tsunami debris (polyethylene, polystyrene, polyvinyl chloride and fibreglass). This unprecedented event confirmed that coastal species can survive in the open ocean for at least six years.

Here we evaluate the extent and composition of living coastal organisms on plastic rafts in the North Pacific Subtropical Gyre (NPSG), thousands of kilometres from the nearest continental margin, in 2018–2019, over seven years after the 2011 tsunami. The goal of our study was to explore the role of floating plastic debris in the transport and persistence of coastal (versus obligate, pelagic) rafting organisms in the open ocean. We aimed to test (1) whether coastal species are common and reproduce in the open ocean on plastic rafts, (2) the extent to which plastic raft type affects the associated community composition, and (3) what similarities and differences in traits exist between the observed coastal and pelagic rafting taxa. Given that plastic tsunami debris with living coastal Japanese taxa continued to land in North America into the spring of 2020 (J.T.C., unpublished observations), albeit rarely, we predicted that a portion of the floating anthropogenic rafts intercepted in the NPSG would be characteristic of Japanese tsunami debris and would still support living coastal species but at lower diversity due to the passage of time[16]. We also predicted that intercepted rafts would most commonly support the obligate, pelagic community, characterized by pelagic barnacles (*Lepas* spp.) and associated neustonic taxa, with relatively few coastal taxa present nearly eight years since the rare tsunami pulse event.

## Results

### Debris and taxa origin

Seventy-five debris items were collected in November 2018, and 30 items were collected in December 2018–January 2019 in the eastern part (east of the dateline) of the NPSG (ENPSG), totalling 105 items across ten designated debris categories (Extended Data Figs. 1 and 2 and Extended Data Table 1). Many items showed advanced degradation commensurate with having been at sea for many years. For example, many plastic bins and baskets, typically a minimum of 3–5 mm in thickness, were now paper-thin and highly friable. Most debris items (85.7%) did not have identifiable markings linked to origin, such as manufacture locations or company/brand names. Eight items (7.6%) bore markings of East Asian origin, including five specifically from Japan. Four debris items (3.8%) had markings commensurate with North American origin. Nearly all taxa were of Northwest Pacific origin, including species from the coast of Japan; representative Japanese taxa included crustaceans (*Ianiropsis serricaudis* and *Jassa marmorata*), sea anemones (*Diadumene lineata*) and bryozoans (*Bugula tsunamiensis*), among others (Supplementary Tables 1 and 2).

### Taxa incidence frequency, richness and accumulation

Invertebrate biofouling was present on 98.0% of debris items. Pelagic species were found on 94.3% of debris items, and coastal species were on 70.5% of debris (Extended Data Fig. 2a). Debris harbouring only pelagic species accounted for 27.6% of all debris, while debris with only coastal species accounted for far less, at 3.8% of items (Extended Data Fig. 2b). Pelagic and coastal taxa were often found together on debris items, with 66.7% of all debris harbouring at least one taxon from each community (Extended Data Fig. 2b). Of the 105 items collected, 103 items were included in the analyses represented below; 2 items were excluded due to the absence of fouling (that is, TOC_MD00013) or the presence of fouling that was excluded from analysis (that is, TOC_MD00016, which only had biofilm attached).

A total of 484 specimens of invertebrate biofouling taxa were collected, comprising 46 taxa from 6 phyla (Supplementary Table 1). Of these taxa, coastal taxa constituted 37 (80%) of the total. Of the six phyla, Bryozoa represented the greatest total richness across all biofouled debris (14 taxa). Arthropoda (Crustacea and Chelicerata) and Cnidaria were also species rich, with 11 and 10 taxa total, respectively (Fig. 1 and Supplementary Table 1). Coastal taxa were more diverse than their pelagic counterparts in all phyla and comprised more than 50% of taxa per taxonomic group (Fig. 1). Crustaceans had the greatest total richness for pelagic taxa (five species) (Fig. 1). In fact, crustaceans were some of the most frequently observed taxa for both communities, representing three of the five most common taxa for both coastal (19.4–36.9% of biofouled debris) and pelagic taxa (25.2–65%; Supplementary Table 1).

A mean (±standard error (s.e.)) of 4.7 (±0.3) taxa was present on biofouled debris. Per debris item, the mean coastal taxa richness was slightly higher (3.1 ± 0.2) than the mean pelagic taxa richness (2.6 ± 0.1), with taxa community (coastal versus pelagic) having a significant effect on taxa richness (generalized linear model (GLM): $F_{1,171} = 4.811$, $P = 0.023$).

Of the ten debris item categories, the mean coastal taxa richness per item was the highest on fishing nets (mean: 4.8 ± 1.2 s.e.), while the mean pelagic taxa richness was the highest on crates (mean: 3.2 ± 0.3 s.e.;

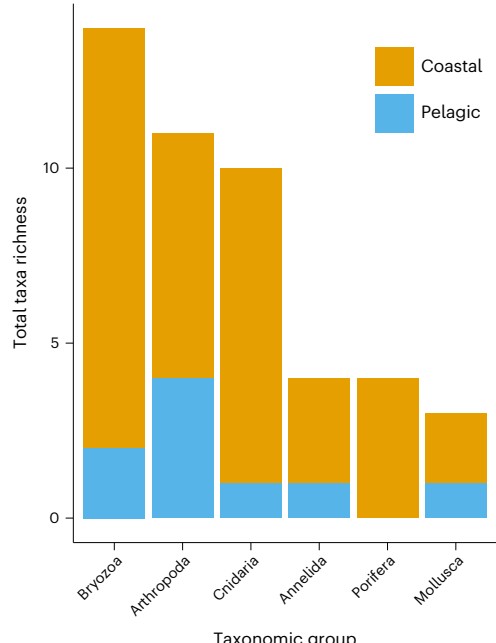

**Fig. 1 | Total taxa richness of biofouling invertebrates per taxonomic group per community (coastal or pelagic) ($n = 103$).** The stacked bars represent the number of taxa of coastal or pelagic communities represented in each phyletic category along the *x* axis. Coastal taxa are depicted in orange. Pelagic taxa are depicted in blue.

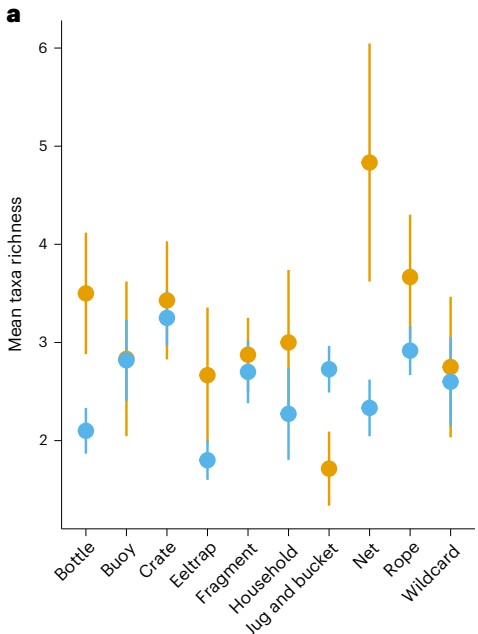

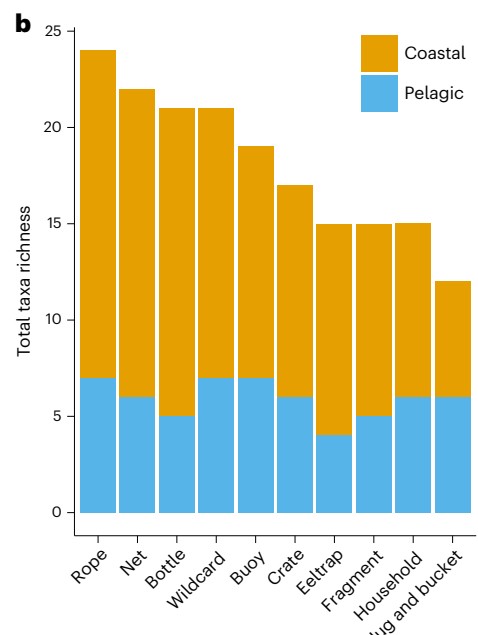

**Fig. 2 | Mean and total taxa richness per debris type for coastal versus pelagic taxa. a**, Mean taxa richness per community (coastal or pelagic) by debris type. **b**, Total taxa richness across debris types per community (coastal or pelagic); the stacked bars represent the number of taxa from coastal or pelagic communities represented in each debris category along the *x* axis. The error bars represent 1 s.e. See Extended Data Fig. 1 or Extended Data Table 1 for the sample sizes per debris type. Coastal taxa are depicted in orange. Pelagic taxa are depicted in blue.

Fig. 2a). Rope had the greatest total taxa richness, harbouring 24 taxa; nets, wildcards (Methods) and bottles also harboured a total of 20 or more taxa (Fig. 2b). Coastal taxa made up most of the total taxa richness per debris type, representing 50–75% of the total invertebrate diversity. However, there was no effect of debris item type on taxa richness (GLM: $F_{9,162}$ = 9.461, $P$ = 0.396), nor was there an interactive effect of debris type and taxa community (GLM: $F_{9,152}$ = 9.376, $P$ = 0.403).

In terms of species accumulation, pelagic taxa richness saturated, reaching an asymptote in taxa accumulation over the 103 biofouled debris analysed, with an asymptotic diversity estimate of 9.495 (±1.312 s.e.) taxa (Fig. 3 and Table 1a). However, coastal taxa richness continued to rise (Fig. 3), with an asymptotic diversity estimate of 47.46 (±7.603 s.e.) taxa (Table 1a), suggesting that saturation in taxa richness has not yet been met. When assessed by debris type, nets had the greatest asymptotic diversity for both coastal and pelagic taxa with 73.75 (±34.914 s.e.) and 7.75 (±3.307 s.e.) taxa, respectively. Overall, similar patterns in coastal versus pelagic communities were observed across debris types (Table 1b), demonstrating that the community-specific patterns were not driven by underlying differences in accumulation by debris type.

### Reproduction and size class structure

We found evidence of reproduction in both pelagic and coastal taxa. Among pelagic taxa, we found ovigerous or brooding females of the crab *Planes* spp. (on 4.9% of biofouled debris) and the caprellid *Caprella andreae* (6.8%). Coastal taxa also showed evidence of reproduction. We observed reproductive structures on the hydroids *Aglaophenia* aff. *pluma* (reproductive on 22.3% of biofouled debris), *Plumularia strictocarpa* (1.9%) and *Antennella secundaria* (1.0%). We also found ovigerous or brooding females of the amphipods *Stenothoe gallensis* (on 5.8% of biofouled debris), *Elasmopus rapax* (1.9%) and *Calliopius pacificus* (1.9%) and of the isopod *Ianiropsis serricaudis* (2.9%).

Evidence of multiple size classes on debris was apparent for both sea anemones and peracarids. On debris with anemones, the average and maximum numbers of size classes present varied by anemone

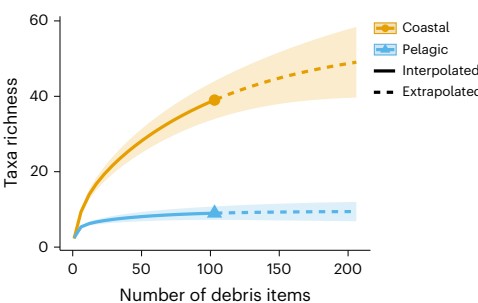

**Fig. 3 | Species accumulation for taxa of each community (coastal or pelagic) across all biofouled debris items ($n$ = 103).** The shaded areas represent the 95% confidence intervals. Coastal taxa richness is depicted in orange. Pelagic taxa richness is depicted in blue. The solid lines represent taxa richness interpolated from presence/absence data, while the dashed lines represent taxa richness extrapolation. The circle and triangle mark the endpoints of interpolated data at 103 debris items for coastal and pelagic taxa, respectively. The species accumulation curves were generated using the iNEXT package in R[47,52,53].

morphotaxa, with *Anthopleura* sp. A having the highest maximum number of size classes (5 classes) and the highest average (2.2 classes) per debris item (Table 2). Of the five coastal amphipod taxa, *Elasmopus rapax*, *Calliopius pacificus* and *Stenothoe gallensis* shared the greatest maximum number of size classes (3 classes), and *Elasmopus rapax* had the greatest average number of size classes (2.0 classes) (Table 2). The coastal isopod *Ianiropsis serricaudis* had an average of 1.2 and a maximum of 3 size classes (Table 2).

### Community analysis

Debris type significantly influenced overall community composition (pelagic and coastal taxa together) in the multivariate community analysis (GLM: residual degrees of freedom (res.df) = 93, residual deviance (dev) = 373.9, $P$ = 0.002, significance level ($\alpha$) = 0.017 for community

**Table 1 | Asymptotic diversity estimates for coastal versus pelagic taxa across all debris items and per debris type**

| (a) All debris | | | | | | |
|---|---|---|---|---|---|---|
| **Taxa community** | | **Taxa richness** | | | | |
| | | Observed | Estimated | s.e. | LCL | UCL |
| Coastal | | 37 | 47.46 | 7.603 | 39.919 | 74.488 |
| Pelagic | | 9 | 9.495 | 1.312 | 9.029 | 17.374 |
| **(b) Per debris type** | | | | | | |
| **Taxa community** | **Debris type** | **Taxa richness** | | | | |
| | | Observed | Estimated | s.e. | LCL | UCL |
| Coastal | Bottle | 16 | 21.760 | 5.384 | 17.219 | 43.221 |
| | Buoy | 12 | 30.409 | 18.398 | 15.603 | 106.056 |
| | Crate | 11 | 13.333 | 3.097 | 11.324 | 27.824 |
| | Eeltrap cone | 11 | 14.750 | 4.437 | 11.599 | 34.468 |
| | Fragment | 10 | 21.250 | 15.462 | 11.494 | 94.699 |
| | Household | 9 | 14.455 | 5.934 | 9.965 | 39.827 |
| | Jug/bucket | 6 | 10.091 | 6.568 | 6.447 | 43.399 |
| | Net | 16 | 73.750 | 34.914 | 35.340 | 188.443 |
| | Rope | 17 | 22.908 | 5.508 | 18.254 | 44.839 |
| | Wildcard | 14 | 32.409 | 18.398 | 17.603 | 108.056 |
| Pelagic | Bottle | 5 | 5.000 | 0.491 | 5.000 | 6.395 |
| | Buoy | 7 | 7.455 | 1.220 | 7.026 | 14.804 |
| | Crate | 6 | 6.000 | 0.522 | 6.000 | 7.628 |
| | Eeltrap cone | 4 | 4.450 | 1.209 | 4.026 | 11.740 |
| | Fragment | 5 | 5.000 | 0.385 | 5.000 | 5.980 |
| | Household | 6 | 6.227 | 0.678 | 6.012 | 10.419 |
| | Jug/bucket | 6 | 6.455 | 1.220 | 6.026 | 13.804 |
| | Net | 6 | 7.750 | 3.307 | 6.156 | 25.607 |
| | Rope | 7 | 7.231 | 0.686 | 7.012 | 11.468 |
| | Wildcard | 7 | 7.000 | 0.577 | 7.000 | 8.767 |

All values were calculated using the iNEXT function with the raw incidence data in the iNEXT package in R[47,52,53]. The 'Observed' column provides the total observed diversity (in this case taxa richness). The 'Estimated' column provides asymptotic estimates of taxa richness, where taxa richness accumulation reaches an asymptote; these estimates were calculated with the ChaoRichness function (see Chao et al.[52] for asymptotic estimator formulas). The 's.e.' column represents the estimated bootstrap standard errors for the asymptotic estimates, and 'LCL' and 'UCL' represent the lower and upper 95% confidence intervals, respectively.

analysis to accommodate multiple analyses), while collection period was a trend (res.df = 92, dev = 66.4, $P$ = 0.020, Extended Data Fig. 3). Univariate species responses were largely non-significant. However, the presence of pelagic bryozoan *Jellyella* spp. and coastal isopod *Ianiropsis serricaudis* were significantly influenced by debris type (GLM: dev = 32.136, $P$ = 0.005 and dev = 32.499, $P$ = 0.005, respectively). When coastal and pelagic communities were analysed separately with multivariate community analyses, we observed differences in response to debris item type and collection period. For the coastal community, the effect of debris type on overall community structure was dampened to a statistical trend (GLM: res.df = 93, dev = 263.99, $P$ = 0.028), while collection period was significant (GLM: res.df = 92, dev = 60.76, $P$ = 0.011). For the pelagic community, debris item type significantly affected overall community structure (GLM: res.df = 93, dev = 109.93, $P$ = 0.010), while sampling period did not (GLM: res.df = 92, dev = 5.68, $P$ = 0.809).

## Life history traits
In addition to the taxonomic richness observed, a diversity of life history traits was present on the sampled debris. We found mild to no correlation between traits (0 to ±0.6) except for mobility and reproduction, which were strongly correlated (0.8) (Extended Data Fig. 4). Taxa incidence frequency was significantly predicted by our overall model, which included taxa community (coastal versus pelagic) and the six life history categories (adult mobility, trophic position, feeding mechanism, reproduction, fertilization and larval development) (GLM: $\chi^2_{15,30}$ = 14.845, $P$ < 0.001). No interactive effect was detected between taxa community and each of the life history categories (GLM: $\chi^2_{6,24}$ = 2.39, $P$ = 0.214).

Our results illustrate that taxa community (coastal or pelagic) was a significant predictor of taxa incidence frequency, with pelagic taxa observed more frequently across debris (GLM: $F_{1,30}$ = 11.522, $P$ = 0.002). For mobility, the mean taxa incidence frequency of pelagic taxa was over three times greater for sessile than for mobile taxa (Fig. 4a); however, mobility was not a significant predictor of taxa incidence frequency (GLM: $F_{1,30}$ = 1.647, $P$ = 0.209). Trophic position did significantly influence taxa incidence frequency (GLM: $F_{3,30}$ = 3.184, $P$ = 0.038), with omnivory almost exclusively observed for pelagic taxa, while all trophic categories were nearly even for coastal taxa (Fig. 4b). Feeding mechanism was also a significant predictor of taxa incidence frequency (GLM: $F_{4530}$ = 3.120, $P$ = 0.022), but the observations were less polarized with a more even spread across suspension feeders, predators, surface feeders and multi-mechanism taxa (Fig. 4c). The remaining life history traits—reproduction, fertilization and larval development—did not significantly influence taxa incidence frequency (GLM: $F_{2,30}$ = 0.015, $P$ = 0.902; $F_{2,30}$ = 0.401, $P$ = 0.674; $F_{2,30}$ = 0.781, $P$ = 0.467, respectively; Fig. 4d–f).

**Table 2 | Size class structure of select Arthropoda (Amphipoda) and Cnidaria (Anthozoa) taxa**

| Phylum | Order/class | Taxon | Taxa community | Mean no. of size classes | Max. no. of size classes | No. of J1 occurrences |
|---|---|---|---|---|---|---|
| Arthropoda | Amphipoda | *Caprella andreae* | Pelagic | 1.4 | 4 | 0 |
| | | *Jassa marmorata* | Coastal | 1.0 | 1 | 0 |
| | | *Elasmopus rapax* | Coastal | 2 | 3 | 2 |
| | | *Stenothoe gallensis* | Coastal | 1.4 | 3 | 1 |
| | | *Calliopius pacificus* | Coastal | 1.2 | 3 | 0 |
| | | Amphilochidae sp. | Coastal | 1.0 | 1 | 0 |
| | Isopoda | *Ianiropsis serricaudis* | Coastal | 1.2 | 3 | 0 |
| Cnidaria | Anthozoa | *Anthopleura* sp. A | Coastal | 2.2 | 5 | 1 |
| | | *Anthopleura* sp. B | Coastal | 1.9 | 3 | 0 |
| | | *Anemonia erythraea** | Coastal | 2 | 4 | 2 |
| | | *Anthopleura* sp. D | Coastal | 1.5 | 2 | 0 |
| | | *Diadumene lineata* | Coastal | 1.7 | 2 | 1 |

Mean and maximum size classes were calculated across the total number of debris items where each taxon occurred. 'No. of J1 occurrences' refers to the number of occurrences of the smallest size class (J1) per taxon: for Arthropoda, J1 < 0.05 mm (from rostrum to telson); for Cnidaria, J1 < 2 mm (at the widest point). *Anthopleura* sp. C was genetically resolved to *Anemonia erythraea*.

Although reproduction and larval development mode did not significantly influence taxa incidence frequency, it is nonetheless noteworthy that 68% of coastal taxa reproduce asexually, whereas 33% of pelagic species do so (Supplementary Table 1). Moreover, while overall coastal and pelagic taxa were similarly composed of non-planktonic (direct or benthic) larval developers (24% and 22%, respectively), three of the top five most frequently observed coastal taxa undergo non-planktonic development (Supplementary Table 1).

## Discussion

In contrast to the long-standing paradigm that coastal taxa largely cannot survive in the open ocean[19,20], coastal species were common and diverse on floating plastic debris in the ENPSG. Generally, invertebrate biofouling was very common, occurring on 103 of the 105 plastic debris items. Pelagic taxa were the dominant community on debris, yet the coastal community was also prevalent, occurring on over 70% of debris. While diversity per item was relatively low, averaging approximately four species per item, overall coastal taxa were more diverse and frequent than anticipated. Moreover, coastal taxa dominated both observed and estimated total richness, as well as observed richness per object type and taxonomic group.

Despite a strong similarity between the coastal species we found on plastic debris in the ENPSG and those previously observed on Japanese tsunami marine debris (JTMD) washed ashore in North America and the Hawaiian Islands (70.3% similarity, Supplementary Table 2)[16], we observed distinct differences in dominant taxonomic groups. For example, Bryozoa and Cnidaria were the most diverse phyletic groups on ENPSG debris, while Mollusca were most diverse on JTMD landings[16]. Indeed, coastal molluscs were largely absent on ENPSG debris, except two bivalve species, *Musculus cupreus* and *Crassostrea gigas*, while JTMD landings rafted over 60 molluscan species from Japan between 2012 and 2017[16]. Of particular interest is the regular abundance (hundreds of individuals on some debris) of sea anemones on ENPSG rafts. Japanese anemones on JTMD were generally sparse from 2012 to 2015, until several major landings of *Diadumene lineata* and *Anthopleura* sp. (neither of which had appeared before) arrived in the spring of 2016 in California, Oregon and Washington. *Anthopleura* sp. again showed up in Oregon in the spring of 2017[21]. Additionally, we found coastal species in the ENPSG not detected on JTMD. These include species of gammarid amphipods and sea anemones, as well as bryozoans (*Amathia gracilis* and *Aetea* spp.). Of interest is the surprisingly common occurrence of the bryozoan *Aetea* sp. A, distinct from that found on JTMD, and whose taxonomic affinities are under investigation.

Overall, species richness was much lower on ENPSG debris (composed of non-JTMD and JTMD debris) than on solely JTMD debris[16]. This may be due to (1) our smaller sample size, (2) the smaller surface area of debris items[22,23], (3) different types of debris (JTMD included docks and vessels)[16] and (4) the biological and environmental constraints of some coastal species being able to survive for long periods in the open ocean. However, the lack of asymptote in our coastal species accumulation curve suggests that coastal taxa richness remains underestimated (Fig. 2), and we can expect to discover more coastal species as marine debris collections continue.

While the diversity and frequency of living coastal taxa found on floating anthropogenic debris in the ENPSG presented here are new, the observation of living coastal species found at sea is not unprecedented. Living coastal species derived from the Western Pacific have been previously documented either in the ENPSG or landing in, for example, North America. These include samples taken at sea, such as a single living specimen of the Asian mussel *Mytilus coruscus* found on a drifting fish net off the coast of Washington in 1986[24] (J.T.C., unpublished observation); mussels (*Mytilus* sp.), Asian oysters and corals found occasionally on buoys since the 1990s (C. Moore, personal communication); and various taxa found on floating marine debris in the ENPSG in 2009 and 2011[22]. Living Western Pacific species landing in North America on marine debris were unknown in the scientific literature prior to studies on JTMD, although beachcomber reports documented rare earlier landings[16].

Our present work stands in contrast to the report by Rech et al.[25] of a relatively low diversity of coastal species in the South Pacific Gyre; however, as Rech et al. note, the diversity of coastal taxa in their study might be higher than estimated. Elsewhere around the world, coastal species have been found rafting in the open ocean or having completed a transoceanic journey and landing on distant shores, often associated with floating kelp and pumice[11,12,14,26–28]. However, in all such cases known to us, the general interpretation of these observations has been that coastal species discovered in the open ocean found themselves in an "unsuitable habitat" and were "misplaced"[22]. That is, neritic taxa were known to be carried out to sea but were long held to be transient and ephemeral in an inhospitable environment whose chemical, biological and physical conditions excluded coastal species from the open ocean[3]. These conclusions were supported by the rarity of reports of coastal biota found surviving on the high seas on long-range, multi-year rafted objects and by lack of evidence of their reproduction in the open ocean.

We also report the discovery of coastal species reproducing on such debris in the open ocean. Size class structure provided further evidence of reproduction. For the amphipods *S. gallensis* and *E. rapax*,

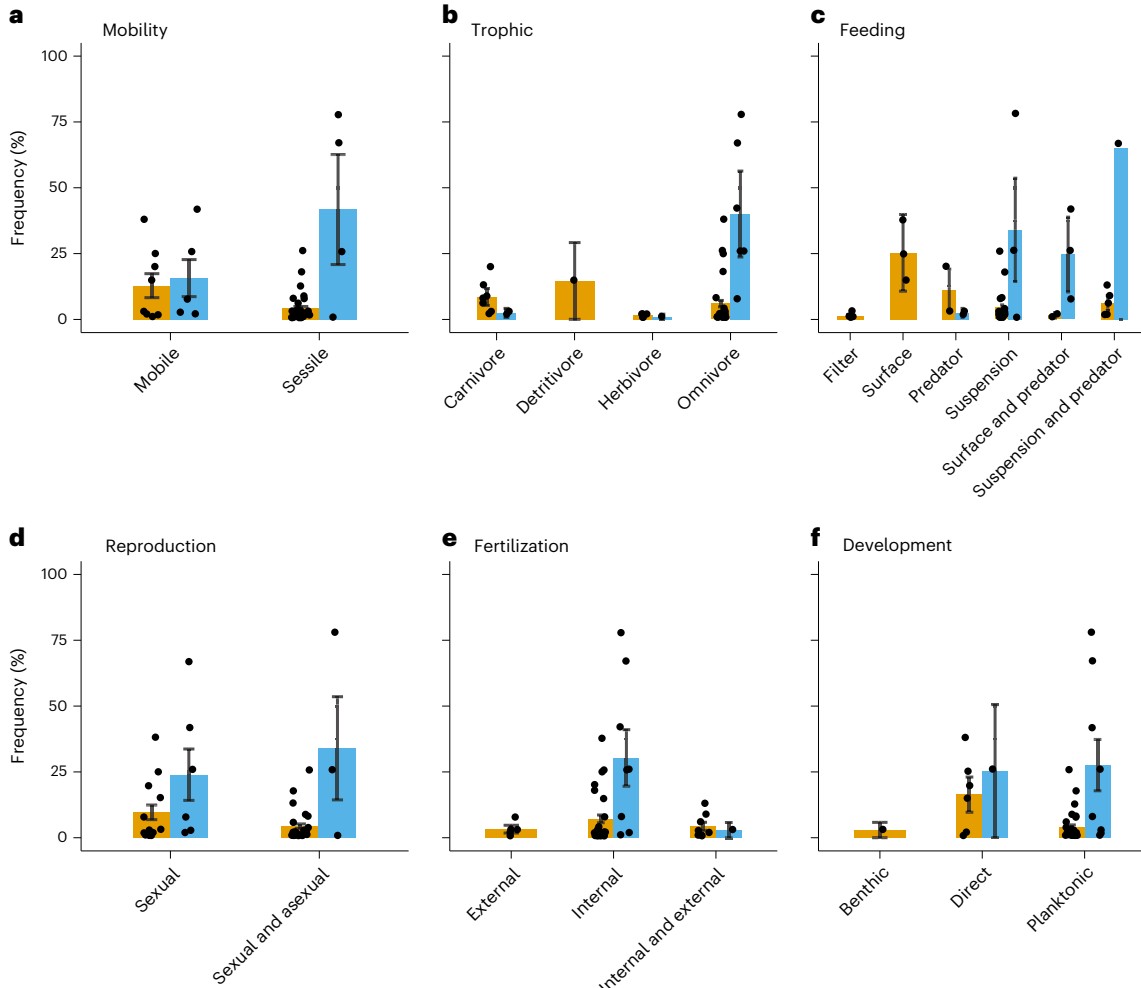

**Fig. 4 | Differences in life history characteristics of coastal versus pelagic biofouling invertebrate communities rafting in the open ocean, depicted as the mean taxa incidence frequency per life history trait present on biofouled debris (n = 103) per community (coastal or pelagic). a**, Adult mobility: sessile or mobile. **b**, Adult diet: carnivore, detritivore, herbivore or omnivore. **c**, Adult feeding mechanism: filter feeder, grazer, predator, suspension feeder, grazer and predator, or suspension feeder and predator. **d**, Reproduction: sexual or sexual and asexual. **e**, Fertilization: external, internal, or internal and external. **f**, Larval development: benthic, direct or planktonic. The error bars represent 1 s.e. See Supplementary Table 2 for the trait details and citations.

individuals less than 0.5 mm long suggest recent reproductive events in the ENPSG (Table 2). Sexual reproduction was also apparent in coastal hydroids, with over 20% of the *A*. aff. *pluma* occurrences displaying reproductive structures. Furthermore, size class distributions among anemone taxa indicated that clonal reproduction took place on debris in the ENPSG, with at least three taxa with new recruits of less than 2 mm. The presence of reproductive females and multiple size classes for coastal crustaceans and cnidarians suggests that coastal species are reproducing and may be self-recruiting and continuously populating their parent rafts in the ENPSG. The top five most frequently observed coastal taxa undergo sexual reproduction with direct development of offspring or clonal, asexual reproduction (Supplementary Table 1). Furthermore, unlike pelagic taxa, most coastal taxa (89%) were characterized by sexual reproduction with non-planktonic larval development, asexual reproduction or both. In fact, the hydroid *A*. aff. *pluma*, which was the most frequently occurring and reproductive coastal species, showed evidence of both sexual and asexual reproduction. Hydroids, in particular, seem to be well suited to surviving and persisting in the transition from coastal to pelagic environments given their varied and flexible morphology as settlers and as epibionts on floating artificial substrates[29,30]. While traits related to reproduction (reproduction mode, fertilization and larval development) did not significantly

influence taxa incidence frequency in our models, these traits may allow for enhanced recruitment either to parent rafts or to adjacent plastic rafts, improving population persistence over long-duration rafting events[2,3,31].

Our results provide insights about novel community dynamics on plastic rafts in the ENPSG. The co-occurrence of pelagic and coastal taxa on 66.7% of biofouled debris, with coastal taxa occurring on 70.5% of debris (either with pelagic taxa or alone), suggests that pelagic and coastal taxa commonly interact and may compete for space and other resources. Further analysis of rafters' traits indicated key predictors of taxa incidence frequency—trophic position and feeding mechanism. Omnivory was the most observed trophic position, while predation, grazing, suspension feeding and combinations of those mechanisms were the most observed feeding mechanisms in this study. These traits may allow the utilization of resources generated by the emergent raft community itself[2]. Previous literature indicates relatively fast development of basal resources, such as algal and bacterial biofilms, on floating plastics[32,33]. Novel nutrient dynamics have also been observed for marine plastics; for example, dissolved organic carbon leachate from low- and high-density polyethylene, polypropylene and polyethylene stimulate marine microbial activity, though leachate and stimulatory effects decrease over time[34]. These emergent properties of plastic rafts

may play an important role in sustaining diverse biofouling communities, but more research is needed to understand how such emergent properties may drive colonization, succession and trophic interactions of coastal and pelagic taxa associated with floating plastics.

Our frequent observations of coastal taxa, consisting of diverse life history traits, age structures and reproductive states, on plastic debris from the ENPSG provide evidence that a standing coastal community, as part of a novel neopelagic community[35], may have established in the open ocean—a phenomenon caused by the introduction of a vast sea of relatively permanent anthropogenic rafts since the 1950s. Rather than coastal organisms as misplaced species in an unsuitable habitat, it may be that substrate availability, and not physiological or ecological constraints, was the critical limiting factor in the exclusion of coastal species from the open ocean[35]. Similarly, the long-lived nature of plastic rafts and the sink-like conditions of the ENPSG[36,37] may allow for the proliferation of different life history traits in the open ocean, as evidenced by the array of traits observed here between pelagic obligate rafters and coastal species (Fig. 4). Thus, the addition of long-term, non-biodegradable rafts in the form of floating plastics, which have increased in collective volume and surface area since the mid-twentieth century[38,39], may be enhancing coastal species survival.

We hypothesize that this neopelagic community is composed of two primary elements: (1) those coastal and pelagic species maintaining self-renewing populations through reproduction, and (2) coastal species that are aperiodically carried out of the coastal zone and can survive for many years but lack the reproductive strategy to become permanent colonists. Species populations in category 2, albeit potentially long-lived, may thus rely on constant infusion of rafting propagules from coastal zones. Category 1 may explain our discovery of abundant sea anemone populations in the ENPSG, while category 2 may explain the concomitant absence of species that were conspicuous elements of JTMD, including the rose barnacle *Megabalanus rosa*, the thatched barnacle *Semibalanus cariosus* and the mussel *Mytilus galloprovincialis*[16].

Our results demonstrate that the oceanic environment and floating plastic habitat are clearly hospitable to coastal species. Coastal species with an array of life history traits can survive, reproduce, and have complex population and community structures in the open ocean. The plastisphere may now provide extraordinary new opportunities for coastal species to expand populations into the open ocean and become a permanent part of the pelagic community, fundamentally altering the oceanic communities and ecosystem processes in this environment with potential implications for shifts in species dispersal and biogeography at broad spatial scales[35]. With plastic pollution waste generation and inputs to the ocean expected to exponentially increase over the next few decades[38,40], a steady source of substrate may sustain the neopelagic as a persistent community. This research presents the cusp of discovery of neopelagic communities, and further innovative and interdisciplinary studies (including of trophodynamics and potential competitive interactions between pelagic and coastal species) are needed to more fully understand the role of floating plastics in ocean gyres[41]. Future research should also investigate the degree to which the patterns observed in the North Pacific Ocean occur in other ocean gyre systems.

## Methods

### Study site

Plastic marine debris collections took place in the ENPSG midway between the coasts of California and Hawaii (Extended Data Fig. 5). The NPSG is an area encircled by the Kuroshio, North Pacific, California and North Equatorial currents where converging surface currents collect a broad variety of floating debris. The ENPSG has been identified as the most heavily plastic-polluted ocean gyre on the globe, currently burdened with at least 79,000 tons of floating plastic debris[39], and the region with the highest debris concentration is thus commonly referred to as the Great Pacific Garbage Patch.

### Marine debris and specimen collections

This study took place as part of a larger research programme known as Floating Ocean Ecosystems (FloatEco), which was funded by the National Aeronautics and Space Administration to investigate the biological and physical underpinnings of the ENPSG through interdisciplinary research and participation of non-profit and volunteer partners, who aided in debris collection (see www.floateco.org for more information about the FloatEco project). For the research presented here, the non-profit the Ocean Cleanup facilitated the collection of floating plastic debris during two expeditions aboard the *Maersk Transporter* in November 2018 and January 2019. Crew members collected floating plastic debris greater than 15 cm in length, width or height. Because of variation in debris type in the ENPSG, we standardized collections by targeting items in ten pre-designated categories based on what is most encountered in that system[39] (L.L., unpublished observation): bottles, buoys, fish crates, eel or hagfish trap cones, plastic fragments, miscellaneous household items (such as clothes hangers and flowerpots), jugs/buckets, fishing nets, rope, and 'wildcard items' (Extended Data Fig. 1). Wildcard items were items that stood out to The Ocean Cleanup observers for their density and diversity of attached biota and did not fit into the other nine designated categories. Approximately ten items were collected in each category (Extended Data Fig. 1). To assess biota frequency on ENPSG debris, items from all categories, except the wildcard category, were collected and sampled as they were encountered. For example, the first ten fishing nets encountered were collected and sampled regardless of evidence of attached biota. We did this to avoid sampling bias towards biota-rich debris. Wildcard items were collected opportunistically throughout the survey period.

Upon collection, each item was given a unique identifying number, and the latitude/longitude, date and time, and debris characteristics (type and size category) were recorded. The item was then photographed from all sides, and it was noted whether biota discernible to the naked eye were attached. If biota appeared to be absent, the item was placed in a plastic bag and frozen or dried on deck if too large for the freezer. If biota were present, either approximately five individuals of each recognizable taxon (morphotype) were removed and preserved in 95% non-denatured ethanol or the entire debris item was frozen for later sampling in the lab. Sampled debris items were placed in individual bags and frozen or dried on the deck. Preserved (frozen, ethanol and dried) items were returned to shore and analysed as follows. Identifying marks (manufacturer names, logos, numbers or locations) were recorded if present. All items were classified into size categories, defined as small (20–50 cm), medium (50–500 cm) or large (>500 cm) (Extended Data Table 1), and examined in a standardized search effort of 15 minutes for macroscopic species, the smaller of which were then examined using a dissecting microscope as needed, outside of the 15-minute search effort. In addition, all frozen and ethanol-stored organisms were preserved or re-preserved in 70% non-denatured ethanol. Voucher specimens from this study are deposited at the Smithsonian Environmental Research Center in Edgewater, Maryland, and the Royal BC Museum in Victoria, British Columbia.

### Taxa analyses

Taxa were identified morphologically (or genetically, in the case of sea anemones) to the lowest possible taxonomic category following the methods outlined in Carlton et al.[16] and with the aid of expert taxonomists ('Author contributions' and 'Acknowledgements'). Species were recorded as present/absent per debris item. Sexual reproductive status for females (gravid versus not gravid) was recorded for decapod and peracarid (amphipod/isopod) crustaceans and hydroids, for which reproductive status is easily identified. Size classes were recorded for sea anemones and peracarids; sea anemones were binned as <2, 2–4, 4–6, 6–10 and >10 mm, and peracarid size classes were binned as <0.5, 0.5–1.5, 1.5–2.5, 2.5–4 and >4 mm. Size classes were selected on the basis

of the size range of individuals present and/or records of juvenile sizes and growth rates for similar taxa based on previous literature (see the JTMD NEMESIS database[42]).

To compare life history traits across coastal and pelagic taxa, we categorized each taxon's adult mobility (mobile or sessile), trophic level (carnivore, detritivore, herbivore or omnivore), feeding mechanism (filter feeder, grazer, predator, suspension feeder or combinations of mechanisms), reproductive strategy (asexual, sexual or sexual/asexual), fertilization strategy (external, internal or internal/external) and larval development type (benthic, direct or planktonic). If life history traits were not well defined for a taxon, we used information available for related higher taxonomic groups. We used life history trait classifications and, when relevant, species life history data from the JTMD NEMESIS database[42].

We focused our analysis on macroinvertebrates; although samples of other taxa (such as algal films and occasional green and red filamentous algae, foraminifera and nematodes) were collected, these were excluded from formal analysis. Macroinvertebrates were relatively common and evenly sampled across debris items, so we had confidence in standard quality for both frequency and community-level analyses. We excluded organisms that were clearly dead prior to preservation (no soft tissue present), which included hydroids with no intact hydranths. Select specimens (including Porifera (sponges) and *Anthopleura* (sea anemones)) were characterized as morphotaxa rather than to species level due to taxonomic identification constraints.

### Categorization of pelagic and coastal taxa

We assigned all taxa to one of two taxa community categories:

(1) Coastal taxa: Species associated with shallow-water, benthic habitats on or shoreward of the inner continental shelf. Included are species typically found in intertidal and sublittoral zones on a wide range of both natural and anthropogenic substrates, the latter including docks, pontoons, piers, pilings and stationary fisheries gear. Examples along the Western and Eastern Pacific coasts include sponges, sea anemones, most hydroids and most bryozoans, bivalves and shelled gastropod molluscs, balanoid barnacles, and ascidians. Taxa that could not be identified to species but were not part of the known pelagic community were considered coastal taxa.

(2) Pelagic taxa: Species typically living as obligate, neustonic organisms in the surface or near-surface waters of the open ocean. Examples in the North Pacific Ocean include the hydroid *Obelia griffini*, the polychaete *Amphinome rostrata*, the caprellid amphipod *Caprella andreae*, the lepadomorph barnacles *Lepas* spp., the crabs *Planes* spp. and *Plagusia* spp., the nudibranch *Fiona pinnata*, and the bryozoans *Jellyella* spp. and *Arbopercula angulata*[16]. These (and similar taxa in other oceans) are believed to have evolved in the high-seas pelagic environment on natural rafts and large mobile fauna, such as whales and sea turtles[43–45].

### Data interpretation and statistical analysis

**Taxa incidence frequency, richness and accumulation.** We first calculated the incidence frequency for the percentage of objects with any coastal taxa, any pelagic taxa, or both coastal and pelagic taxa when considering all collected debris ($n = 105$). As noted earlier, two items had either no fouling or biofilms only. For the biofouled debris ($n = 103$), we then described the mean and total taxa richness per debris type (see Supplementary Table 1 for the sample sizes per debris type), both for taxa overall and per taxa community (coastal versus pelagic). We also calculated the incidence frequency (%) of each taxon on the collected biofouled debris ($n = 103$). We calculated and visualized these descriptive statistics using the tidyverse, dplyr, stats and ggplot2 packages in R v.3.6.2 (refs. [46–49]).

To determine the effect of taxa community (coastal versus pelagic) and debris item type on taxa richness, we analysed taxa richness as a function of taxa community and debris item type, including the interaction between the two predictor variables, using a GLM with a Poisson distribution in R using the MASS package[47,50]. We also included sampling period as a fixed effect in this model to account for variation in sampling effort between the two sampling periods (12–22 November 2018 and 16 December 2018 to 3 January 2019), as more debris items were collected in period 1. However, sampling period was not a significant predictor variable and did not improve model fit, so it was removed. We also performed a Tukey post-hoc test to detect pairwise differences in the effects of origin and item type on taxa richness (AER package[51]). Statistical significance was assessed at $\alpha = 0.05$, and statistical trends were assessed at $0.05 < \alpha < 0.10$.

To determine the saturation of coastal versus pelagic taxa richness on all debris and on each debris type, we constructed taxa accumulation curves using the iNEXT package in R[47,52,53]. iNEXT uses taxa incidence data to interpolate accumulation curves across samples on the basis of observed data and can be used to extrapolate the accumulation curve for up to twice the number of samples observed. To identify differences in taxa accumulation between the coastal and pelagic communities, we used the iNEXT function to create an accumulation curve across our total sample size of 103 biofouled debris and then extrapolate the taxa richness accumulation up to 206 debris items. We used the same method to identify differences in accumulation between coastal and pelagic communities for each debris category. Sample sizes, and thus interpolation/extrapolation potential, varied among debris categories (see Extended Data Fig. 1 and Extended Data Table 1 for the sample sizes). In addition to producing species accumulation curves, we used the iNEXT function to calculate the asymptotic diversity estimate, or the taxa richness value at which point the accumulation curve reaches an asymptote, for each of these analyses; these estimates were calculated using the ChaoRichness function embedded within the iNEXT function (see Chao et al.[52] for asymptotic estimator formulas).

**Reproduction and size class structure.** We assessed the incidence frequency (%) of reproductive individuals on biofouled debris ($n = 103$) for hydrozoan and crustacean taxa ('Taxa analyses') of each community. We also assessed the maximum and mean (±s.e.) number of size classes observed per biofouled debris ($n = 103$) for anemone and peracarid taxa. As above, these descriptive statistics were calculated and visualized in R[46–49].

**Community analysis.** To determine the effect of debris characteristics on community structure, we analysed species composition as a function of debris type (see Extended Data Fig. 1 and Extended Data Table 1 for the debris type factor levels and replication), with collection period as a blocking factor to help control for variation possibly introduced during the two expeditions (taxa incidence ~ debris type + collection period). Because the community data were collected as taxa presence/absence, we used a multivariate GLM with binomial family, logit link and Montecarlo resampling in the mvabund package in R[54] following the methods of Wang et al.[55]. P values were generated using the default 999 iterations via PIT-trap resampling. The mvabund multivariate approach is a model-based analysis of the overall community structure with simultaneous univariate analysis per species[55]. This statistical method is superior to distance-based methods of community analysis (for example, non-metric multidimensional scaling, principal component analysis and permutational multivariate analysis of variance) for datasets with rarity and zero-inflation, as is the case in our dataset[55,56]. In addition to this full-community analysis, we separated taxa on the basis of their taxa community (coastal or pelagic) and ran a multivariate community analysis (following the same procedure as detailed above) for each origin to gain a higher-resolution understanding of differences between the two communities. To account for resampling the dataset

for three separate analyses, we used a Bonferroni correction, evaluating statistical significance using an adjusted $\alpha$ value ($\alpha$ = 0.050/3 = 0.017). Community structure was visualized with non-metric multidimensional scaling plots (with Bray–Curtis differences) created using the vegan and ggplot2 packages in R[46,47,57].

We also descriptively investigated the overall phyletic diversity observed. To do so, we calculated the total number of phyla observed and the taxa richness for each. We also separated taxa richness for each phylum by taxa community (coastal versus pelagic). As above, we calculated and visualized these descriptive statistics in R[46–49].

**Life history traits.** Finally, we sought to determine whether the open-ocean environment favoured taxa with specific life history traits and whether coastal taxa present in the open ocean were characterized by similar life history traits as the observed pelagic taxa. To do so, we calculated the incidence frequency (%) for each coastal and pelagic taxon across biofouled debris. Each taxon was assigned a life history trait within six life history categories: adult mobility, trophic level, feeding mechanism, reproduction, fertilization and larval development (Table 2; see 'Marine debris and specimen collections' for a description of the trait assignment methodology). We then analysed taxa incidence frequency as a function of taxa community (coastal versus pelagic) and the six life history trait categories using a GLM. Because taxa incidence frequency data are positive and continuous, bounded between 0 and 1, we square-root-transformed the data and applied a gamma distribution with an inverse link. For this analysis, taxa incidence frequencies were estimated across collection periods, and thus we did not include collection period as a predictor variable in the model. We did not have a priori expectations that the life history traits studied would have combined, interactive effects on taxa incidence frequency. However, we wanted to know if interactive effects exist between taxa community (coastal versus pelagic) and each of the life history traits. Our model thus assessed taxa incidence frequency as a function of the interactive effect of taxa community and each life history trait (taxa incidence frequency ~ taxa community × (mobility + trophic level + feeding mechanism + reproduction + fertilization + larval development). We compared the model that included the interaction of taxa community and life history traits with a reduced model without the interaction using a likelihood ratio test. We note that the life history variables explored here may be correlated; to account for this, we assessed correlations between the life history variables. We found one instance of high correlation between mobility and reproduction but maintained both variables in the model because they provide important and different information about the life history strategies of observed taxa. The data were analysed and visualized using the MASS, GGally, ggplot2, tidyverse and dplyr packages in R[46–50,58].

**Study site plastic concentration model**
With the details of sea-based and land-based sources of debris not known, simulations of marine debris pathways and patterns is very difficult. Exceptions are the so-called garbage patches, areas in the subtropical oceans where converging surface currents produce high concentrations of floating debris[59]. After being trapped in these patches for a long time, debris items (and model particles) 'forget' their origin, which allows the development of a model of the garbage patch insensitive to source distribution supplying the patch with new model debris. We used SCUD (Surface Currents from Diagnostic[60]), an empirical model of the surface currents derived from the trajectories of drifters drogued at 15 m (see Centurioni[61] for a recent description of the technology) deployed by the Global Drifter Program[62–64] (see also https://gdp.ucsd.edu/ldl/global-drifter-program/) and collocated wind and geostrophic currents derived from satellite products. This model was used in previous studies of the Indian Ocean debris pathways from general sources[65] and after the MH370 flight disappearance[66] and demonstrated better agreement with reports from North America

and Hawaii of debris generated by the 2011 tsunami in Japan than other ocean circulation models, such as HYCOM (https://www.hycom.org/)[59].

The model was forced by the QuikSCAT satellite winds from 2000 to 2009 and by the ASCAT satellite winds from 2010 to 2018, intercalibrated with QuikSCAT during an overlap of more than one year between the missions. Numerical experiments started with no tracer in the ocean and proceeded with the sources uniformly distributed along the global coastline. To reduce the effect of the initial condition, we looped the model in time, and to sustain an equilibrium solution, we added dissipation with an $e$-fold timescale of ten years. Before the beginning of the new cycle, a one-year 'sponge' period was added with currents and winds linearly interpolated between 2018 (in the previous cycle) and 2000 (in the new cycle). The difference between tracer concentrations in the end of the third and fourth model loops was less than 3%. The final solution was then projected into the future, and additional experiments confirmed its relatively weak sensitivity to the model setup.

### Inclusion and ethics statement
No research ethics approvals were required for this work. All authors included in this paper were essential to the research and writing process. No researchers were excluded from the publication.

### Reporting summary
Further information on research design is available in the Nature Portfolio Reporting Summary linked to this article.

### Data availability
The data for this research are available at the Dryad data depository: https://doi.org/10.5061/dryad.k98sf7m9d.

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

## Acknowledgements

The FloatEco project (www.floateco.org) was funded in part by the US National Aeronautics and Space Administration (grant no. 80NSSC17K0559) through membership in Biodiversity, Ecological Forecasting and Ocean Surface Topography Science Teams. The samples used in this study were collected by collaborators at The Ocean Cleanup during their 2018–2019 expeditions aboard the *Maersk Transporter*, funded by The Ocean Cleanup donors. We thank J. Chapman, N. Hitchcock, L. McCann and T. Phillips for their expertise and assistance with the identification of rafting marine invertebrates. We thank the captains and crew of the *Maersk Transporter*, L. Khatmullina from The Ocean Cleanup and N. Treneman for assistance with debris collection; J. Crooks and the team at the Tijuana River National Estuarine Research Reserve for help with sample logistics; J. Troubaugh for lab assistance; and M. Lonneman for help with data analysis. We also thank C. Moore for communicating his earlier observations of corals and bivalve molluscs on buoys in the ENPSG.

## Author contributions

L.E.H., J.T.C., L.C., M.C., J.H., V.H., N.M., C.M., G.M.R., A.S. and C.W. are core members of the FloatEco research group. L.E.H., J.T.C., C.M. and G.M.R. designed the study, and L.C., M.C., J.H., N.M., A.S., C.W. and V.H. provided input during implementation and analysis. L.E.H., M.E. and L.L. coordinated the study logistics and sample collection. M.E. collected the samples. L.E.H., J.T.C., J.P., H.C., B.C. and M.M. analysed the samples. L.E.H. analysed the results. L.E.H., J.T.C. and G.M.R. wrote the paper. J.P., L.C., M.C., J.H., N.M., A.S., C.W., V.H., M.E., L.L., H.C., B.C. and M.M. contributed to the paper.

## Competing interests

M.C., a member of the FloatEco team, directs Ocean Voyages Institute (https://www.oceanvoyagesinstitute.org/), which is a non-profit organization, headquartered in Sausalito, California, that conducts marine debris cleanup operations in the ENPSG, among other areas across the globe, and works with members of the FloatEco research team to improve our understanding of physical drivers of plastic accumulation in the ENPSG and its ramifications for marine biodiversity. L.L. and M.E. are employed by The Ocean Cleanup (https://theoceancleanup.com/), a non-profit organization aimed at advancing scientific understanding and developing solutions to rid the oceans of plastic, headquartered in Rotterdam, the Netherlands. The remaining authors of this manuscript have no competing interests to disclose.

## Additional information

**Extended data** is available for this paper at https://doi.org/10.1038/s41559-023-01997-y.

**Correspondence and requests for materials** should be addressed to Linsey E. Haram.

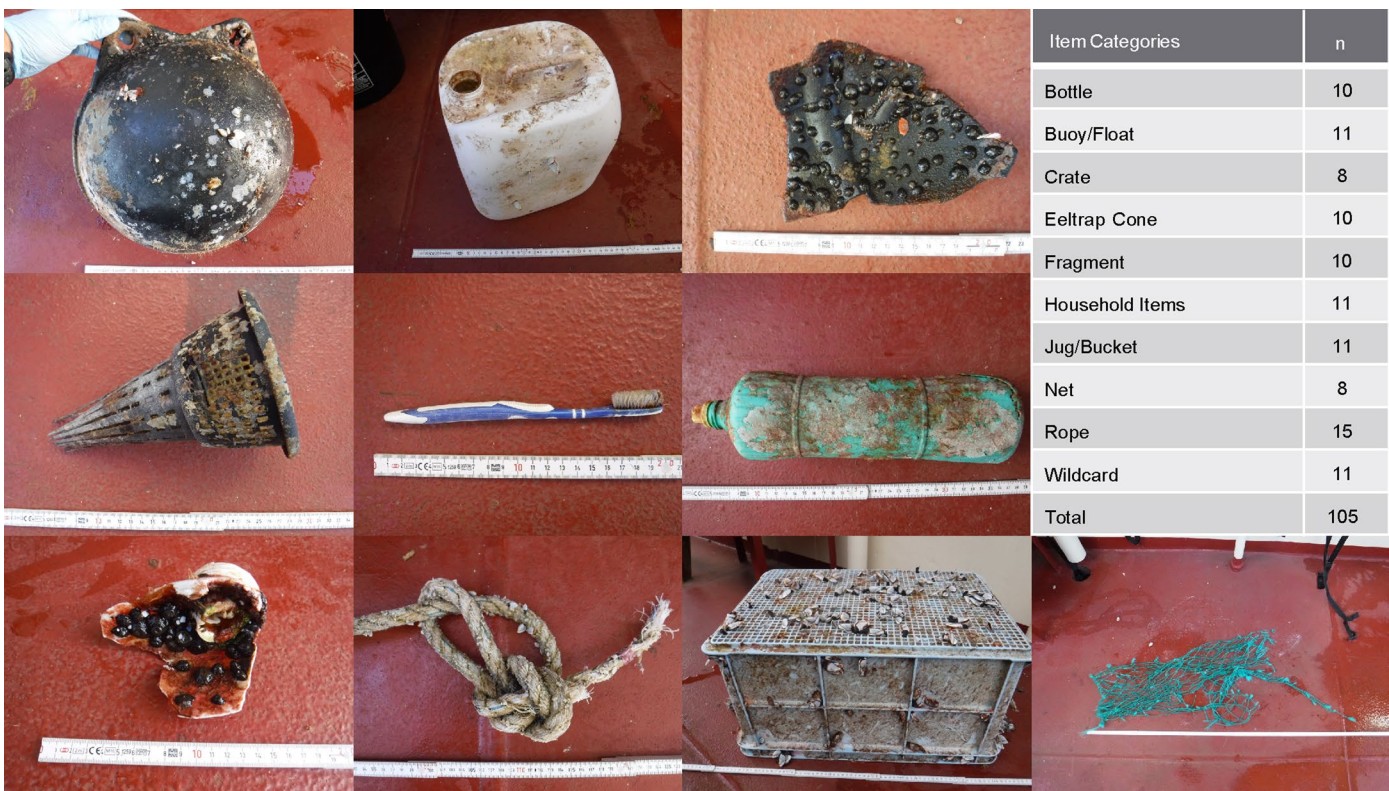

| Item Categories | n |
| --- | --- |
| Bottle | 10 |
| Buoy/Float | 11 |
| Crate | 8 |
| Eeltrap Cone | 10 |
| Fragment | 10 |
| Household Items | 11 |
| Jug/Bucket | 11 |
| Net | 8 |
| Rope | 15 |
| Wildcard | 11 |
| Total | 105 |

**Extended Data Fig. 1 | Marine debris categories (or types) collected for analysis from the Eastern North Pacific Subtropical Gyre.** From top left: Buoy/float, Jug/bucket, Fragment, Eeltrap cone, Household items, Bottle, Wildcard, Rope, Crate, and Net. Inset table depicts the sample size (#) per category. Photos courtesy of The Ocean Cleanup.

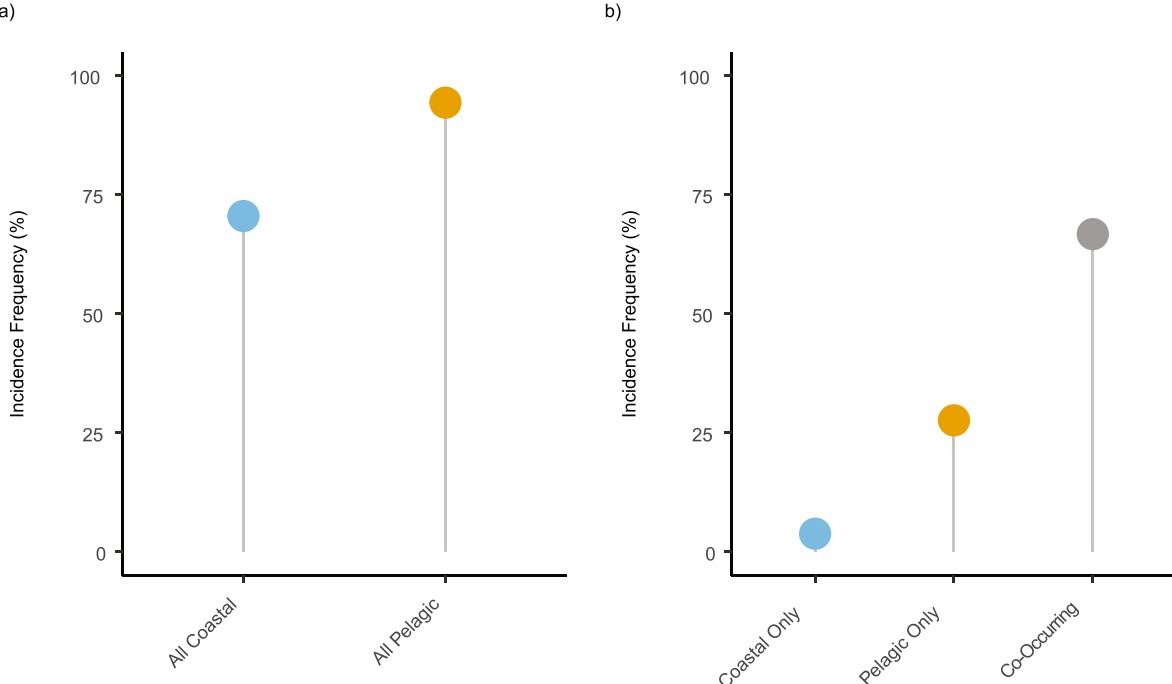

**Extended Data Fig. 2 | Incidence frequency (%) of biofouling invertebrates on all collected plastic debris (n = 105).** Incidence frequency (%) of biofouling macroinvertebrates on plastic debris (n = 105) for items with (a) coastal or pelagic taxa, or items with (b) only coastal, only pelagic, or both community types co-occurring.

a) Debris Item Type

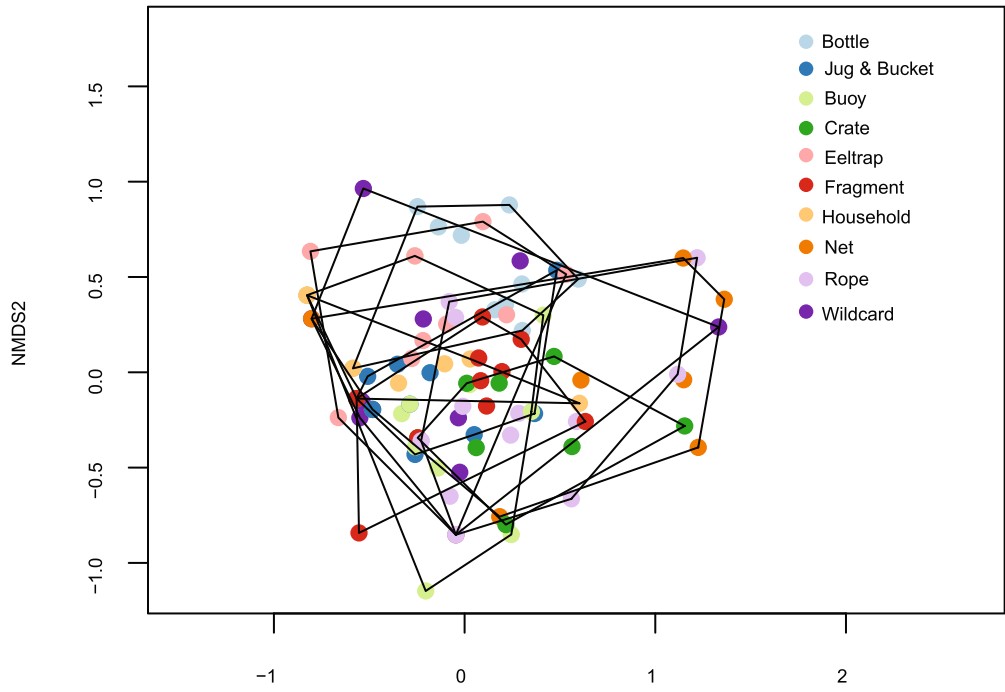

b) Collection Period

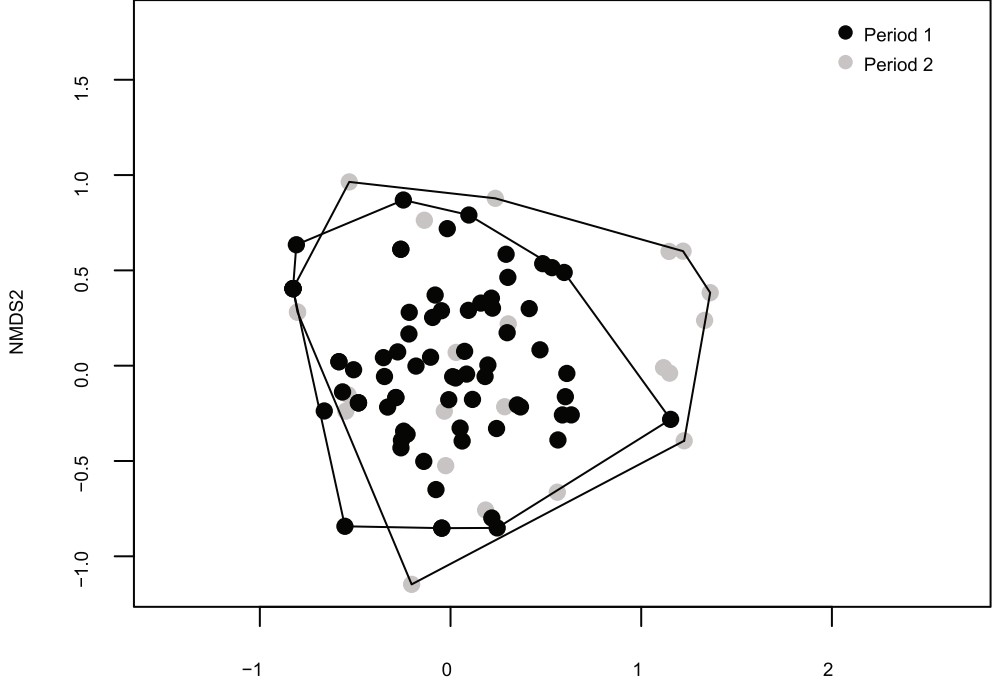

**Extended Data Fig. 3 | Community composition of biofouling invertebrates across all debris item (n=103) by a) debris type and b) collection period.** nMDS representation of community composition by (a) debris type and (b) collection period for the entire biofouling invertebrate community, including all coastal and pelagic taxa. Two debris items without biofouling were excluded for an overall n = 103 (see Table S1 for sample sizes per debris type and size). Stress = 0.145, k = 3.

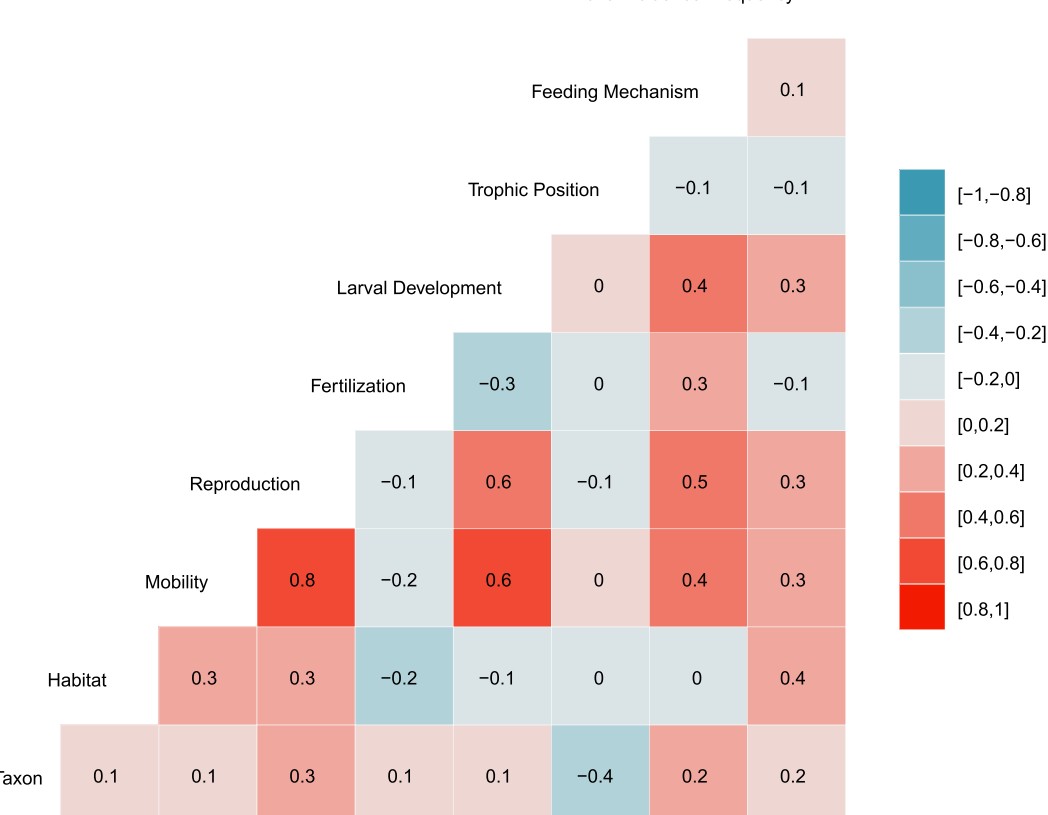

**Extended Data Fig. 4 | Correlation of life history traits and taxa incidence frequency observed on plastic marine debris.** Each square represents the correlation between the variable denoted as the row label and that denoted by the column label. For example, the topmost square in the far right column depicts the correlation (0.1) between 'feeding mechanism' and 'taxa incidence frequency'. Correlations range from −1 to 1. Zero represents no correlation, while 1 and −1 represents strong positive and strong negative correlations, respectively. The blue spectrum represents the range of negative correlations from 0 to −1 (light to dark blue). The red spectrum represent the range of positive correlations from 0 to 1 (pink to dark red).

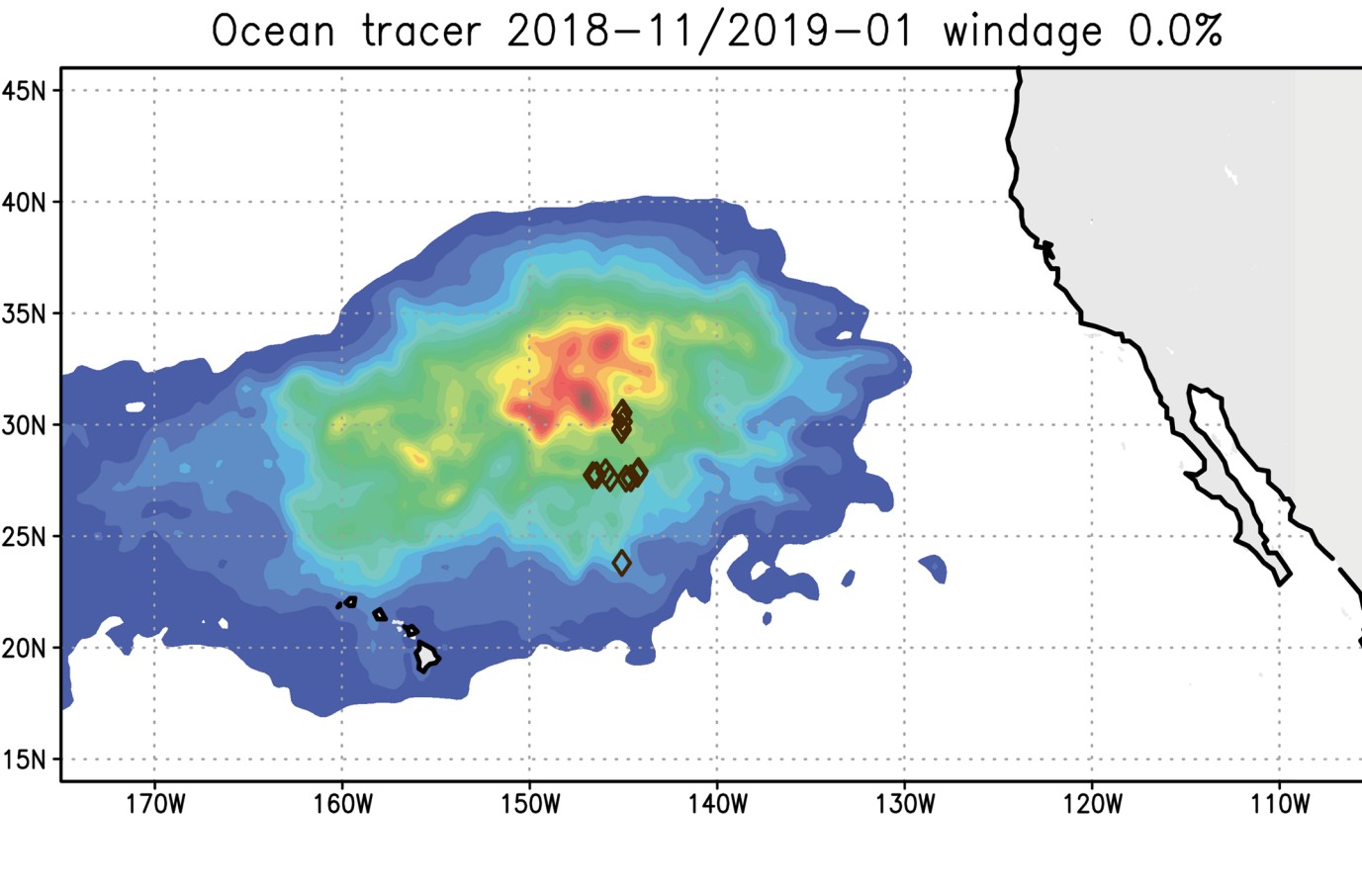

**Ocean tracer 2018−11/2019−01 windage 0.0%**

**Extended Data Fig. 5 | Collection sites of floating plastic debris within the Eastern North Pacific Ocean Subtropical Gyre, or 'North Pacific Garbage Patch'.** Debris collection sites, illustrated as diamonds, in the Eastern North Pacific Ocean Subtropical Gyre. The underlying ocean tracer map illustrates the predicted concentration of debris within the study area during the time of collection, assuming 0% windage of debris, with 0 representing low concentrations of debris and 1 representing high concentrations of debris.

**Extended Data Table 1 | Number of debris items collected per debris category across size categories**

| Debris Category | Count | S | M | L |
|---|---|---|---|---|
| Bottle | 10 | 10 | 0 | 0 |
| Buoy/Float | 11 | 9 | 2 | 0 |
| Crate | 8 | 4 | 4 | 0 |
| Eeltrap Cone | 10 | 10 | 0 | 0 |
| Fragment | 10 | 8 | 2 | 0 |
| Household | 11 | 11 | 0 | 0 |
| Jug/Bucket | 11 | 10 | 1 | 0 |
| Net | 8 | 0 | 8 | 0 |
| Rope | 15 | 6 | 9 | 0 |
| Wildcard | 11 | 7 | 3 | 1 |
| Total | 105 | 75 | 29 | 1 |

Count = number of items collected per debris category; S = small; M = medium; L = large.

# Reporting Summary

## Statistics

For all statistical analyses, confirm that the following items are present in the figure legend, table legend, main text, or Methods section.

| n/a | Confirmed | |
|---|---|---|
| ☐ | ☒ | The exact sample size (*n*) for each experimental group/condition, given as a discrete number and unit of measurement |
| ☐ | ☒ | A statement on whether measurements were taken from distinct samples or whether the same sample was measured repeatedly |
| ☐ | ☒ | The statistical test(s) used AND whether they are one- or two-sided *Only common tests should be described solely by name; describe more complex techniques in the Methods section.* |
| ☐ | ☒ | A description of all covariates tested |
| ☐ | ☒ | A description of any assumptions or corrections, such as tests of normality and adjustment for multiple comparisons |
| ☐ | ☒ | A full description of the statistical parameters including central tendency (e.g. means) or other basic estimates (e.g. regression coefficient) AND variation (e.g. standard deviation) or associated estimates of uncertainty (e.g. confidence intervals) |
| ☐ | ☒ | For null hypothesis testing, the test statistic (e.g. *F*, *t*, *r*) with confidence intervals, effect sizes, degrees of freedom and *P* value noted *Give P values as exact values whenever suitable.* |
| ☐ | ☒ | For Bayesian analysis, information on the choice of priors and Markov chain Monte Carlo settings |
| ☐ | ☒ | For hierarchical and complex designs, identification of the appropriate level for tests and full reporting of outcomes |
| ☒ | ☐ | Estimates of effect sizes (e.g. Cohen's *d*, Pearson's *r*), indicating how they were calculated |

*Our web collection on statistics for biologists contains articles on many of the points above.*

## Software and code

Policy information about availability of computer code

| Data collection | Analyses were conducted using standard methods for the statistical packages used in R. |
|---|---|
| Data analysis | R Studio |

For manuscripts utilizing custom algorithms or software that are central to the research but not yet described in published literature, software must be made available to editors and reviewers. We strongly encourage code deposition in a community repository (e.g. GitHub). See the Nature Portfolio guidelines for submitting code & software for further information.

## Data

Policy information about availability of data

All manuscripts must include a data availability statement. This statement should provide the following information, where applicable:
- Accession codes, unique identifiers, or web links for publicly available datasets
- A description of any restrictions on data availability
- For clinical datasets or third party data, please ensure that the statement adheres to our policy

The data for this research are available upon publication at the Dryad data depository indicated in the manuscript. https://doi.org/10.5061/dryad.k98sf7m9d

## Human research participants

Policy information about studies involving human research participants and Sex and Gender in Research.

| | |
|---|---|
| Reporting on sex and gender | The present study did not involved human research participants. |
| Population characteristics | *Describe the covariate-relevant population characteristics of the human research participants (e.g. age, genotypic information, past and current diagnosis and treatment categories). If you filled out the behavioural & social sciences study design questions and have nothing to add here, write "See above."* |
| Recruitment | *Describe how participants were recruited. Outline any potential self-selection bias or other biases that may be present and how these are likely to impact results.* |
| Ethics oversight | *Identify the organization(s) that approved the study protocol.* |

Note that full information on the approval of the study protocol must also be provided in the manuscript.

# Field-specific reporting

Please select the one below that is the best fit for your research. If you are not sure, read the appropriate sections before making your selection.

☐ Life sciences  ☐ Behavioural & social sciences  ☒ Ecological, evolutionary & environmental sciences

For a reference copy of the document with all sections, see nature.com/documents/nr-reporting-summary-flat.pdf

# Ecological, evolutionary & environmental sciences study design

All studies must disclose on these points even when the disclosure is negative.

| | |
|---|---|
| Study description | This study took place as part of a larger research program known as Floating Ocean Ecosystems (FloatEco), which was funded by the National Aeronautics and Space Administration to investigate the biological and physical underpinnings of the ENPSG through interdisciplinary research and participation of non-profit and volunteer partners, who aided in debris collection (see www.floateco.org for more information about the FloatEco project). For the research presented here, the non-profit The Ocean Cleanup facilitated collection of floating plastic debris during two expeditions aboard Maersk Transporter in November 2018 and January 2019. |
| Research sample | Crewmembers collected floating plastic debris greater than 15 cm in length, width, or height. |
| Sampling strategy | Because of variation in debris type in the ENPSG, we standardized collections by targeting items in 10 pre-designated categories based on what is most encountered in that system41 (L. Lebreton, personal observation): bottles, buoys, fish crates, eel or hagfish trap cones, plastic fragments, miscellaneous household items (such as clothes hangers and flowerpots), jugs/buckets, fishing nets, rope, and "wildcard items" (Supplementary Fig. S2). Wildcard items were items that stood out to observers for their density and diversity of attached biota and/or did not fit into the other 9 designated categories. In total, approximately 10 items were collected within each category (Fig. S1). To assess biota frequency on ENPSG debris, items from all categories except the wildcard category were collected and sampled as they were encountered. For example, the first 10 fishing nets encountered were collected and sampled regardless of evidence of attached biota. We did this to avoid sampling bias toward biota-rich debris. Wildcard items were collected opportunistically throughout the survey period. |
| Data collection | Upon collection, each item was given a unique identifying number and the latitude/longitude, date and time, and debris characteristics (type, size category) were recorded by shipboard personnel. Subsequently, the item was photographed from all sides and it was noted if biota discernible to the naked eye were attached. If biota appeared to be absent, the item was placed in a plastic bag and frozen or dried on deck if too large for the freezer. If biota were present, either approximately 5 individuals of each recognizable taxon (morphotype) were removed and preserved in 95% non-denatured ethanol or the entire debris item was frozen for later sampling in the lab. Sampled debris items were placed in individual bags and frozen or dried on the deck. Preserved (frozen, ethanol, and dried) items were returned to shore and analyzed as follows. Identifying marks (manufacturer names, logos, numbers, or locations) if present were recorded. All items were classified into size categories, defined as small (20-50 cm), medium (50-500 cm), or large (> 500 cm) (see Supplementary Table S1), and examined in a standardized search effort of 15 minutes for macroscopic species, the smaller of which were then examined using a dissecting microscope as needed, outside of the 15 minute search effort. In addition, all frozen and ethanol organisms were preserved or re-preserved in 70% non-denatured ethanol. Voucher specimens from this study are deposited at the Smithsonian Environmental Research Center in Edgewater, Maryland and the Royal BC Museum in Victoria, British Columbia. |
| Timing and spatial scale | There were two sampling periods: the two sampling periods: November 12-22, 2018 and December 16, 2018 - January 3, 2019. |
| Data exclusions | No data were excluded from the analyses. |
| Reproducibility | No experiments were conducted. |

| Randomization | All samples collected were analyzed. |
| Blinding | All samples collected were analyzed; no blinding was required. |

Did the study involve field work?    ☒ Yes    ☐ No

## Field work, collection and transport

| Field conditions | Ocean-going vessels sailed onto the high seas to collect samples. |
| Location | Collections took place in the Eastern North Pacific Ocean Subtropical Gyre (ENPSG) midway between the coasts of California and Hawaii (Supplementary Fig. S1). The North Pacific Subtropical Gyre is an area of convergence encircled by the Kuroshio, North Pacific, California, and North Equatorial Currents. In the eastern part of the North Pacific Subtropical Gyre (east of the dateline), converging surface currents collect floating debris. |
| Access & import/export | The samples were collected on the high seas beyond any national jurisdiction. No import-export permits or other permits were required in this work. |
| Disturbance | No disturbances were caused or created by this study; by removing plastic, we were helping to minimize or remove disturbance. |

# Reporting for specific materials, systems and methods

We require information from authors about some types of materials, experimental systems and methods used in many studies. Here, indicate whether each material, system or method listed is relevant to your study. If you are not sure if a list item applies to your research, read the appropriate section before selecting a response.

### Materials & experimental systems

| n/a | Involved in the study |
|---|---|
| ☒ | ☐ Antibodies |
| ☒ | ☐ Eukaryotic cell lines |
| ☒ | ☐ Palaeontology and archaeology |
| ☐ | ☒ Animals and other organisms |
| ☒ | ☐ Clinical data |
| ☒ | ☐ Dual use research of concern |

### Methods

| n/a | Involved in the study |
|---|---|
| ☒ | ☐ ChIP-seq |
| ☒ | ☐ Flow cytometry |
| ☒ | ☐ MRI-based neuroimaging |

## Animals and other research organisms

Policy information about studies involving animals; ARRIVE guidelines recommended for reporting animal research, and Sex and Gender in Research

| Laboratory animals | No laboratory animals were involved. |
| Wild animals | The study did not involve wild animals. |
| Reporting on sex | Not applicable. |
| Field-collected samples | If invertebrate biota were present, either approximately 5 individuals of each recognizable taxon (morphotype) were removed and preserved in 95% non-denatured ethanol or the entire debris item was frozen for later sampling in the lab. Sampled debris items were placed in individual bags and frozen or dried on the deck. Preserved (frozen, ethanol, and dried) items were returned to shore and analyzed as follows. Identifying marks (manufacturer names, logos, numbers, or locations) if present were recorded. All items were classified into size categories, defined as small (20-50 cm), medium (50-500 cm), or large (> 500 cm) (see Supplementary Table S1), and examined in a standardized search effort of 15 minutes for macroscopic species, the smaller of which were then examined using a dissecting microscope as needed, outside of the 15 minute search effort. In addition, all frozen and ethanol organisms were preserved or re-preserved in 70% non-denatured ethanol. Voucher specimens from this study are deposited at the Smithsonian Environmental Research Center in Edgewater, Maryland and the Royal BC Museum in Victoria, British Columbia. |
| Ethics oversight | No ethical approval or guidance was required. No human subjects were involved and no animals were subjected to field or laboratory experiments. |

Note that full information on the approval of the study protocol must also be provided in the manuscript.

