## [Peer Review File · Nature Ecology & Evolution]

Peer Review Information

Journal: Nature Ecology & Evolution

Manuscript Title: Extent and reproduction of coastal species in the North Pacific Subtropical Gyre on plastic debris

Corresponding author name(s): Linsey E. Haram

Editorial Notes:

Reviewer Comments & Decisions:

Decision Letter, initial version:

24th June 2022

Dear Dr Haram,

Thank you for your patience while we sought referee feedback on your manuscript entitled "Formation of Coastal Communities on the High Seas: Extent and reproduction of coastal species on plastic in the North Pacific Subtropical Gyre", and apologies for the length of time it has taken to convey this decision to you. Unfortunately, 2 referees withdrew themselves part way through the review process, which caused some delays. However, I can now confirm that it has been seen by 2 reviewers, whose comments are attached. The reviewers have raised a number of concerns which will need to be addressed before we can offer publication in Nature Ecology & Evolution. We will therefore need to see your responses to the criticisms raised and to some editorial concerns, along with a revised manuscript, before we can reach a final decision regarding publication.

As you can see from the reports, the referees comments are relatively minor and should be straightforward to address.

In addition to this, please address the following Editorial comments:

- Please edit the title slightly. We do not permit punctuation in titles, which should be brief and informative. For example "Extent and reproduction of coastal species on plastic in the North Pacific Subtropical Gyre" would be fine.
- Please look through our policy on Competing Interests (<https://www.nature.com/nature-portfolio/editorial-policies/competing-interests>) and include any relevant declarations. In particular, for maximum transparency we suggest also including a link to the list of Ocean Cleanup Foundation donors in this section.
- Please reduce the number of display items (Figures and Tables) to a maximum of 6. For example, Figs 1 and 2 could be moved to a Supplementary Information file.

We therefore invite you to revise your manuscript taking into account all reviewer and editor comments. Please highlight all changes in the manuscript text file in Microsoft Word format.

2When revising your manuscript:

* If you have not done so already please begin to revise your manuscript so that it conforms to our Article format instructions at <http://www.nature.com/natecolevol/info/final-submission>. Refer also to any guidelines provided in this letter.

[REDACTED]

Nature Ecology & Evolution is committed to improving transparency in authorship. As part of our efforts in this direction, we are now requesting that all authors identified as 'corresponding author' on published papers create and link their Open Researcher and Contributor Identifier (ORCID) with their account on the Manuscript Tracking System (MTS), prior to acceptance. ORCID helps the scientific community achieve unambiguous attribution of all scholarly contributions. You can create and link your ORCID from the home page of the MTS by clicking on 'Modify my Springer Nature account'. For more information please visit please visit www.springernature.com/orcid.

[REDACTED]

Reviewers' comments:

Reviewer #1 (Remarks to the Author):

I found this a really interesting and well-written paper on the kinds of coastal fouling species that are found on plastic debris in the North Pacific Subtropical Gyre and factors that might be contributing to these patterns. I don't have any major concerns with the manuscript, but was hoping that the authors might be able to respond to a few minor points that I was intrigued by while reading the paper.

P3 Ln17. Is another possible scenario that the materials are eaten by larger fish or mammals? Or is that included in (2)

P9 Ln23. I found it interesting that invertebrates dominated the biofouling and only an algal film is mentioned (P18, Ln 15). Were there no macroalgae colonising the plastic debris and I was hoping there might be space to comment on this invertebrate dominance in the discussion.

P13 Ln6. While reading about the traits that might be supporting survival and persistence on plastic rafts, I was wondering if the coastal species identified in this study had members that are known to be invasive/have been introduced to new areas via biofouling.

P15 Ln10. Could you expand on the innovative and interdisciplinary research that is needed? I'd also like to know a bit more about the other ocean gyre systems and whether they would be expected to support similar communities.

P17 Ln12. Could you provide a little more justification for a 15 minute search effort. What proportion of the community did that sample and would you expect a bias against any groups of species?

P19 Ln18. Was there any obvious reason why two of the debris items were unfouled?

Reviewer #2 (Remarks to the Author):

P1, L22 onwards to P3, L4: the authors discussion of the abundance, size and role of pumice is too brief. Without more evidence, this seems too big an assumption: that most pumice is small, perhaps not especially abundant, relatively ephemeral in a marine context, and cannot support much biodiversity. Perhaps the latter is true, from personal experience, I rarely observe goose barnacles or biofilm on pumice, but the authors provide minimal evidence to support this. Much of the beach-washed pumice I've witnessed is actually quite large in size, particularly in relation to plastic debris items on the same beaches. For example, see the images included in these recent articles <https://7news.com.au/weather/environment/massive-pumice-raft-washing-up-on-beaches-could->

3help-with-great-barrier-reef-recovery--c-1256063

<https://www.stuff.co.nz/environment/82679002/storms-unveil-giant-pumice-boulders-and-condominium-along-kapiti-coast-beach>

<https://www.goldcoastbulletin.com.au/news/gold-coast/pumice-strewn-across-coast-beaches-was-caused-by-an-underwater-volcano-eruption--about-18-months-ago/news-story/df1b7163590d801a11d751d241240504>

Huge rafts of pumice have recently been recorded. While I'm not aware of estimates of the quantities present in our oceans, the 2012 Havre eruption produced 400 km² of pumice in a single day. Note that most pieces were 5–15 cm in diameter with 1000s of individual pieces in excess of 75cm, which is quite large; DOI 10.1038/ncomms4660)

P3, L20-22: "dispersal of coastal organisms on either natural or anthropogenic rafts resulting in continental landing has rarely been observed and documented. An example of one such rare event occurred when millions of objects were swept into the North Pacific Ocean due to the Great East Japan Tsunami". Please also see Barnes, D.K.A., Fraser, P.P., 2003. Rafting by five phyla on man-made flotsam in the Southern Ocean. *Marine Ecology Progress Series* 262, 289-291.

P10, L10-22: these unusual landings in CA, WA, etc are intriguing – can you provide an explanation as to why this has occurred in recent years, but otherwise hasn't been documented? What is the likely mechanism, or major change that's taken place?

P11, L1-7: this paragraph summarizes your findings for species richness, but again, doesn't provide much insight into the underlying mechanism and/or consequences of what you're reporting.

P11, L8: "evidence of reproduction of living coastal taxa found on floating anthropogenic debris in the ENPSG presented here is novel". This seems to be an important finding, but the next full page of text deviates from this topic and the authors never seem to loop back and discuss the implications. For example, on P5, L15 you state "coastal species were on 70.5% of debris" and on P9, L15 indicate it's "noteworthy that 68% of coastal taxa reproduce asexually compared to 33% for pelagic species". Given coastal species were so widespread and clearly capable of breeding in the locations which you found them, what does this mean for biodiversity? For example, one of the species you recorded (*Stenothoe gallensis*) is well-known for invading new environments. This is mentioned, but indirectly & very briefly, on P15, L7. Given the abundance and diversity of Category 1 coastal species you detected, I believe this warrants further discussion.

P12, L9: ah, I see reproduction by coastal sp. is addressed on this page making some (but not all) of my comments above irrelevant. Please note line 9 on P11 and P12 are somewhat redundant, this contributed to the confusion.

*****END*****

Response to Reviewers 1 and 2

We thank both reviewers for their comments leading to improvements in our paper.

Reviewer #1 (Remarks to the Author):

I found this a really interesting and well-written paper on the kinds of coastal fouling species that are found on plastic debris in the North Pacific Subtropical Gyre and factors that might be contributing to these patterns. I don't have any major concerns with the manuscript, but was hoping that the authors might be able to respond to a few minor points that I was intrigued by while reading the paper.

P3 Ln17. Is another possible scenario that the materials are eaten by larger fish or mammals? Or is that included in (2)

Thank you for this comment. Predation is captured in both scenarios (2) and (3). We have added a sentence relative to the possibility of predation.

P9 Ln23. I found it interesting that invertebrates dominated the biofouling and only an algal film is mentioned (P18, Ln 15). Were there no macroalgae colonizing the plastic debris and I was hoping there might be space to comment on this invertebrate dominance in the discussion.

Thank you. We found occasional green filamentous and red filamentous algae, and have now noted that as well on page 18. We do not exclude the possibility that some species of larger coastal macroalgae may be able to adapt and reproduce in the open ocean environment, but we are hesitant to speculate about this at this time (given, for example, the wide range of algal reproductive strategies (including asexual reproduction, such as found in weedy colonizing species of *Codium*), as to their absence in the present sample set.

P13 Ln6. While reading about the traits that might be supporting survival and persistence on plastic rafts, I was wondering if the coastal species identified in this study had members that are known to be invasive/have been introduced to new areas via biofouling.

5Thank you. The introduction history of species in the Northeast Pacific was covered in Supplementary Table S3. Species known to have introduction histories are specified in the “Notes and References” section of the Table.

P15 Ln10. Could you expand on the innovative and interdisciplinary research that is needed?

We have added a phrase with examples of such research, including trophodynamics and experimental work on competitive interactions.

I'd also like to know a bit more about the other ocean gyre systems and whether they would be expected to support similar communities.

We commented on this in the companion work, Haram et al. (2021): “An important early step is to determine whether neopelagic communities like those found in the North Pacific form in other oceans, and if so, to what extent these communities differ among ocean basins“, and did not feel we needed to repeat that statement here.

P17 Ln12. Could you provide a little more justification for a 15 minute search effort. What proportion of the community did that sample and would you expect a bias against any groups of species?

Thank you. A standardized search effort reduces investigatory bias in terms of investing more time in some objects than in others. We found that after 10 to 15 minutes we detected and sampled the macroscopic species present, observing no morphologically distinct macroinvertebrates beyond this duration. For clarity, as we do not report microscopic species such as protists, we have edited the sentence to focus on microscopic analysis of smaller macroscopic species, which analysis was outside the 15 minute time range.

P19 Ln18. Was there any obvious reason why two of the debris items were unfouled?

Thank you. In the main text, we noted that two of the 105 items had (1) no biofouling or (2) biofilm development only. We have added that clarification again in Methods. The history of debris items can be complex, and it is often not possible to know why certain items may have developed no or little macrofouling.

Reviewer #2 (Remarks to the Author):

P1, L22 onwards to P3, L4: the authors discussion of the abundance, size and role of pumice is too brief. Without more evidence, this seems too big an assumption: that most pumice is small, perhaps not especially abundant, relatively ephemeral in a marine context, and cannot support much biodiversity. Perhaps the latter is true, from personal experience, I rarely observe goose barnacles or biofilm on pumice, but the authors provide minimal evidence to support this. Much of the beach-washed pumice I've witnessed is actually quite large in size, particularly in relation to plastic debris items on the same beaches. For example, see the images included in these recent articles

<https://7news.com.au/weather/environment/massive-pumice-raft-washing-up-on-beaches-could-help-with-great-barrier-reef-recovery--c-1256063>

<https://www.stuff.co.nz/environment/82679002/storms-unveil-giant-pumice-boulders-and-condom-tin-along-kapiti-coast-beach>

<https://www.goldcoastbulletin.com.au/news/gold-coast/pumice-strewn-across-coast-beaches-was-caused-by-an-underwater-volcano-eruption--about-18-months-ago/news-story/df1b7163590d801a11d751d241240504>

Huge rafts of pumice have recently been recorded. While I'm not aware of estimates of the quantities present in our oceans, the 2012 Havre eruption produced 400 km² of pumice in a single day. Note that most pieces were 5–15 cm in diameter with 1000s of individual pieces in excess of 75cm, which is quite large; DOI 10.1038/ncomms4660)

We thank Reviewer 2 for noting that the story of pumice rafting may be complex, and we have modified our wording accordingly. As our focus in this paper is on the presence of coastal species in the North Pacific Ocean, our goal in these few introductory sentences is to acknowledge that we, and the rafting science community in general, are aware that floating vegetation and pumice are known to carry and transport species, but, critically, for far shorter periods of time than anthropogenic debris. We emphasize that our concern here is both a temporal component (our cited literature demonstrates shorter at sea longevity of

7pumice) and an area-diversity component (citing the key literature), but we have added that the total known diversity of species on pumice (but not per capita, which we underscore) may be rich. Importantly, our paper is not the venue to launch into a discussion of pumice as a dispersal mechanism (which is also not an important vector in the North Pacific Ocean), and we feel that if we were to significantly expand this discussion, we would be obligated to similarly discuss at length the role of floating vegetation as well, which encompasses an even larger literature. Since this is the Introduction to the paper, our aim was to highlight what is known about coastal species over long-periods, to provide context for the current study results.

P3, L20-22: “dispersal of coastal organisms on either natural or anthropogenic rafts resulting in continental landing has rarely been observed and documented. An example of one such rare event occurred when millions of objects were swept into the North Pacific Ocean due to the Great East Japan Tsunami”. Please also see Barnes, D.K.A., Fraser, P.P., 2003. Rafting by five phyla on man-made flotsam in the Southern Ocean. *Marine Ecology Progress Series* 262, 289-291.

We know this paper well (Barnes and Fraser 2003). The paper presents evidence that local Antarctic species colonized Antarctic-sourced debris, and does not apply to the system or spatial scales that we describe in the present paper.

P10, L10-22: these unusual landings in CA, WA, etc are intriguing – can you provide an explanation as to why this has occurred in recent years, but otherwise hasn’t been documented? What is the likely mechanism, or major change that’s taken place?

We are similarly intrigued by these later pulses of species, but have little evidence in hand at this time by which to speculate on the drivers of observed differences in taxonomic composition from that observed in 2016-2017 in the Japanese tsunami marine debris communities.

P11, L1-7: this paragraph summarizes your findings for species richness, but again, doesn’t provide much insight into the underlying mechanism and/or consequences of what you’re reporting.

Thank you. We feel that here we have identified 4 distinct underlying mechanisms that may contribute to the differences in R in our current work versus JTMD. Given the novelty of what we are reporting – we emphasize in our conclusions that this “research presents the cusp of discovery of neopelagic communities” – we feel that further speculation is premature.

P11, L8: “evidence of reproduction of living coastal taxa found on floating anthropogenic debris in the ENPSG presented here is novel”. This seems to be an important finding, but the next full page of text deviates from this topic and the authors never seem to loop back and discuss the implications. For example, on P5, L15 you state “coastal species were on 70.5% of debris” and on P9, L15 indicate it’s “noteworthy that 68% of coastal taxa reproduce asexually compared to 33% for pelagic species”. Given coastal species were so widespread and clearly capable of breeding in the locations which you found them, what does this mean for biodiversity? For example, one of the species you recorded (*Stenothoe gallensis*) is well-known for invading new environments. This is mentioned, but indirectly & very briefly, on P15, L7. Given the abundance and diversity of Category 1 coastal species you detected, I believe this warrants further discussion.

As the reviewer notes in their next comment below, we continue to interpret on page 12 the importance of reproductive strategies. Importantly, we are, again, at the frontier of these discoveries, and hesitate to launch into extended speculation on what the presence of these many coastal species means for changes in open ocean diversity over time.

P12, L9: ah, I see reproduction by coastal sp. is addressed on this page making some (but not all) of my comments above irrelevant. Please note line 9 on P11 and P12 are somewhat redundant, this contributed to the confusion.

Thank you. We have altered the wording on page 11 to reduce the redundancy on page 12.

Decision Letter, first revision:

20th September 2022

9Dear Dr. Haram,

Thank you for submitting your revised manuscript "Extent, reproduction, and persistence of coastal species in the North Pacific Subtropical Gyre on plastic debris" (NATECOLEVOL-220516450A). It has now been seen again by the original reviewers. Referee #1 was satisfied with the revisions and had no further comments to pass on. Referee #2 also had no further comments that they wished to pass on formally, however, they have let us know that they felt disappointed with the response to their comments regarding the influence of pumice in this system, which they felt had been dismissed.

As the reviewers find that the paper has improved in revision, we are therefore happy in principle to publish it in Nature Ecology & Evolution, pending minor revisions to comply with our editorial and formatting guidelines, and to address Referee #2's final requests to add more discussion regarding the role of pumice.

In particular, please include some discussion regarding how the small size of pumice particles may limit colonisation but plastics may not, even though the majority of plastics in the ocean are microplastics. In addition, please include some further discussion of pumice as a potential dispersal mechanism, and cite Barnes & Fraser (2003) MEPS 262:289-291 as a relevant previous paper.

[REDACTED]

Our ref: NATECOLEVOL-220516450A

28th September 2022

10Dear Dr. Haram,

Thank you for your patience as we've prepared the guidelines for final submission of your Nature Ecology & Evolution manuscript, "Extent, reproduction, and persistence of coastal species in the North Pacific Subtropical Gyre on plastic debris" (NATECOLEVOL-220516450A). Please carefully follow the step-by-step instructions provided in the attached file, and add a response in each row of the table to indicate the changes that you have made. Please also check and comment on any additional marked-up edits we have proposed within the text. Ensuring that each point is addressed will help to ensure that your revised manuscript can be swiftly handed over to our production team.

****We would like to start working on your revised paper, with all of the requested files and forms, as soon as possible (preferably within two weeks). Please get in contact with us immediately if you anticipate it taking more than two weeks to submit these revised files.****

In recognition of the time and expertise our reviewers provide to Nature Ecology & Evolution's editorial process, we would like to formally acknowledge their contribution to the external peer review of your manuscript entitled "Extent, reproduction, and persistence of coastal species in the North Pacific Subtropical Gyre on plastic debris". For those reviewers who give their assent, we will be publishing their names alongside the published article.

Nature Ecology & Evolution offers a Transparent Peer Review option for new original research manuscripts submitted after December 1st, 2019. As part of this initiative, we encourage our authors to support increased transparency into the peer review process by agreeing to have the reviewer comments, author rebuttal letters, and editorial decision letters published as a Supplementary item. When you submit your final files please clearly state in your cover letter whether or not you would like to participate in this initiative. Please note that failure to state your preference will result in delays in accepting your manuscript for publication.

Cover suggestions

As you prepare your final files we encourage you to consider whether you have any images or illustrations that may be appropriate for use on the cover of Nature Ecology & Evolution.

11We accept TIFF, JPEG, PNG or PSD file formats (a layered PSD file would be ideal), and the image should be at least 300ppi resolution (preferably 600-1200 ppi), in CMYK colour mode.

Nature Ecology & Evolution has now transitioned to a unified Rights Collection system which will allow our Author Services team to quickly and easily collect the rights and permissions required to publish your work. Approximately 10 days after your paper is formally accepted, you will receive an email in providing you with a link to complete the grant of rights. If your paper is eligible for Open Access, our Author Services team will also be in touch regarding any additional information that may be required to arrange payment for your article.

Please note that *Nature Ecology & Evolution* is a Transformative Journal (TJ). Authors may publish their research with us through the traditional subscription access route or make their paper immediately open access through payment of an article-processing charge (APC). Authors will not be required to make a final decision about access to their article until it has been accepted. [Find out more about Transformative Journals](https://www.springernature.com/gp/open-research/transformative-journals)

Authors may need to take specific actions to achieve [compliance with funder and institutional open access mandates](https://www.springernature.com/gp/open-research/funding/policy-compliance-faqs). If your research is supported by a funder that requires immediate open access (e.g. according to [Plan S principles](https://www.springernature.com/gp/open-research/plan-s-compliance)) then you should select the gold OA route, and we will direct you to the compliant route where possible. For authors selecting the subscription publication route, the journal's standard licensing terms will need to be accepted, including [those licensing terms will supersede any other terms that the author or any third party may assert apply to any version of the manuscript](https://www.nature.com/nature-portfolio/editorial-policies/self-archiving-and-license-to-publish).

[REDACTED]

[REDACTED]

Reviewer #2:

None

Reviewer #3:

None

Author Rebuttal, first revision:

NATECOLEVOL-220516450A_Haram

Response to Reviewer #2 re: Revised Manuscript

Re:

address Referee #2's final requests to add more discussion regarding the role of pumice. In particular, please include some discussion regarding how the small size of pumice particles may limit colonisation but plastics may not, even though the majority of plastics in the ocean are microplastics. In addition, please include some further discussion of pumice as a potential dispersal mechanism, and cite Barnes & Fraser (2003) MEPS 262:289-291 as a relevant previous paper.

We thank the reviewer for inquiring about pumice again.

In the revision, we felt that simply removing the specific discussion of pumice characteristics was more efficient, logical, and reasonable, rather than expanding the discussion about pumice.

13Were we to expand the pumice discussion (as the reviewer suggests), most of the entire very first paragraph of the paper would be focused on a phenomenon, vector, and subject that was not the focus of the research we are reporting. A full and proper discourse comparing across a multicharacter field plastic debris with pumice would be the subject of a different paper. Importantly, there would be no scholarly reason to delve into greater details of pumice without going into further comparative details of other natural vectors.

The key difference between plastic debris and natural vectors is object longevity: Plastic debris floats across oceans for a decade or more. Natural objects do not last half that long at most, and almost always for a much smaller fraction of the time.

We emphasize this key point and provide citations, and the opening now reads as follows (citations omitted here:)

Rafting, or the association of organisms with floating debris, has been an inferred mode of marine species dispersal since the 19th century. Yet empirical evidence of floating debris' role in long-term, transoceanic rafting of coastal marine species is limited. The importance of coastal species dispersal by open ocean rafting may depend largely on the nature of the raft material. Natural rafts consist of buoyant, floating vegetation or pumice (the buoyant rock formed during volcanic eruptions). Natural materials are relatively short lived, decomposing at sea over a matter of months or a few years, becoming waterlogged and sinking, or being biodegraded or consumed by marine animals.

What then follows is a comparison to non-biodegradable, long-lasting, plastic.

Re:

“In particular, please include some discussion regarding how the small size of pumice particles may limit colonisation but plastics may not, even though the majority of plastics in the ocean are microplastics.”

We have now removed size comparisons and comparative richness (diversity) loads.

That said, we appreciate the remark here by the Reviewer, though we were studying only macroplastics.

Re:

and cite Barnes & Fraser (2003) MEPS 262:289-291 as a relevant previous paper.

We have now cited Barnes and Fraser.

Final Decision Letter:

24th January 2023

Dear Dr Haram,

We are pleased to inform you that your Article entitled "Extent and reproduction of coastal species in the North Pacific Subtropical Gyre on plastic debris", has now been accepted for publication in Nature Ecology & Evolution.

Over the next few weeks, your paper will be copyedited to ensure that it conforms to Nature Ecology and Evolution style. Once your paper is typeset, you will receive an email with a link to choose the appropriate publishing options for your paper and our Author Services team will be in touch regarding any additional information that may be required

You will not receive your proofs until the publishing agreement has been received through our system

Due to the importance of these deadlines, we ask you please us know now whether you will be difficult to contact over the next month. If this is the case, we ask you provide us with the contact information (email, phone and fax) of someone who will be able to check the proofs on your behalf, and who will be available to address any last-minute problems . Once your paper has been scheduled for online publication, the Nature press office will be in touch to confirm the details.

15Acceptance of your manuscript is conditional on all authors' agreement with our publication policies (see www.nature.com/authors/policies/index.html). In particular your manuscript must not be published elsewhere and there must be no announcement of the work to any media outlet until the publication date (the day on which it is uploaded onto our web site).

Please note that *Nature Ecology & Evolution* is a Transformative Journal (TJ). Authors may publish their research with us through the traditional subscription access route or make their paper immediately open access through payment of an article-processing charge (APC). Authors will not be required to make a final decision about access to their article until it has been accepted. [Find out more about Transformative Journals](https://www.springernature.com/gp/open-research/transformative-journals)

Authors may need to take specific actions to achieve [compliance with funder and institutional open access mandates](https://www.springernature.com/gp/open-research/funding/policy-compliance-faqs). If your research is supported by a funder that requires immediate open access (e.g. according to [Plan S principles](https://www.springernature.com/gp/open-research/plan-s-compliance)) then you should select the gold OA route, and we will direct you to the compliant route where possible. For authors selecting the subscription publication route, the journal's standard licensing terms will need to be accepted, including [those licensing terms will supersede any other terms that the author or any third party may assert apply to any version of the manuscript](https://www.nature.com/nature-portfolio/editorial-policies/self-archiving-and-license-to-publish).

We welcome the submission of potential cover material (including a short caption of around 40 words) related to your manuscript; suggestions should be sent to Nature Ecology & Evolution as electronic files (the image should be 300 dpi at 210 x 297 mm in either TIFF or JPEG format). Please note that such pictures should be selected more for their aesthetic appeal than for their scientific content, and that colour images work better than black and white or grayscale images. Please do not try to design a cover with the Nature Ecology & Evolution logo etc., and please do not submit composites of images

16related to your work. I am sure you will understand that we cannot make any promise as to whether any of your suggestions might be selected for the cover of the journal.

You can generate the link yourself when you receive your article DOI by entering it here: <http://authors.springernature.com/share>.

[REDACTED]

P.S. Click on the following link if you would like to recommend Nature Ecology & Evolution to your librarian <http://www.nature.com/subscriptions/recommend.html#forms>

** Visit the Springer Nature Editorial and Publishing website at http://editorial-jobs.springernature.com?utm_source=ejP_NEcoE_email&utm_medium=ejP_NEcoE_email&utm_campaign=ejP_NEcoE for more information about our career opportunities. If you have any questions please click [here](mailto:editorial.publishing.jobs@springernature.com). **